# BioDynaSpec: Harmonic-Guided Spatio-Spectral Autoregressive Diffusion for Protein Dynamics Generation

**Mujie Lin** [* 1 2]  **Yutian Liu** [* 3]  **Yudi Guo** [* 1 2]  **Yanzhen Hou** [4 5]  **Yiheng Tao** [1 6]  **Ruochong Zheng** [1 2]
**Kaiwen Cheng** [1 2]  **Xin Shan** [1]  **Youdong Mao** [2 5 7 8 9 †]  **Jie Chen** [1 2 6 10 †]

## Abstract

Generating long-horizon molecular dynamics (MD) is difficult due to error accumulation in time-domain autoregressive models, which causes drift, and fixed step-size constraints on temporal resolution. We propose BioDynaSpec, which reformulates protein dynamics as spatio-spectral generation: Independent Windowed Fourier Decomposition (IWFD) decomposes trajectories into window-wise spectral representations, and a generator combines low-to-high frequency autoregression with diffusion denoising to reconstruct continuous motion. This formulation is motivated by a local near-equilibrium view of protein dynamics: after per-window alignment, fluctuations around an anchor conformation are better characterized in spectral space, where local mode structure is more explicit than in framewise coordinates. To improve cross-residue and cross-frequency consistency, we introduce Inter-Residue Frequency Coupling (IRFC), a learnable Gaussian distance bias in attention that injects a resonance-inspired structural prior. On ATLAS, BioDynaSpec improves 250-frame trajectory generation with $R_{250} = 1.509$ Å, where

$R_s$ denotes the mean per-frame C$\alpha$-RMSE over the first $s$ frames after alignment, reducing error by 60.4% versus MDGEN and 57.2% versus ProAR, while achieving the best PCA-2D displacement-profile correlation and stepwise distribution matching. For equilibrium conformational sampling, it achieves Root Mean $W_2 = 1.31$, MD PCA $W_2 = 0.90$, and Joint PCA $W_2 = 1.19$, improving over the next best method by 50.03%, 35.25%, and 47.58%, respectively. It also improves near-equilibrium local-dynamics and covariance consistency, achieving PCA-PSD-LogCorr $= 0.817$ and CFRE $= 0.989$, corresponding to a 21.9% gain and a 36.7% reduction over the next best method, respectively. The source code is available at `https://github.com/Linmj-Judy/BioDynaSpec.git`.

## 1. Introduction

Proteins are dynamic macromolecules that undergo continuous conformational fluctuations to execute essential biological functions, ranging from enzymatic catalysis to signal transduction (Kokkinidis et al., 2012). While experimental techniques such as X-ray crystallography and Cryo-EM provide high-resolution structural information (Fenwick, 2021), they are often limited to discrete snapshots or ensemble-averaged distributions (DeVore & Chiu, 2022). Consequently, resolving the continuous kinetics and transient pathways of short-lived, high-energy transitions remains a formidable challenge (Šrajer & Schmidt, 2017). Molecular Dynamics (MD) simulations offer a computational bridge to explore these dynamics at atomic resolution (McCammon et al., 1977). However, classical MD is constrained by the *timescale gap*: the necessity of femtosecond integration steps ($10^{-15}$ s) for numerical stability renders the sampling of biologically relevant timescales (milliseconds to seconds) computationally prohibitive for most systems (Shaw et al., 2008; Abraham et al., 2015).

To address these limitations, Deep Generative Models (DGMs) have emerged as an efficient alternative for sam-

---

[*]Equal contribution ,[†]Corresponding Author, [1]School of Electronic and Computer Engineering, Peking University, Shenzhen, China [2]AI for Science (AI4S)-Preferred Program, Peking University Shenzhen Graduate School, China [3]School of Computer Science, Peking University, Beijing, China [4]State Key Laboratory for Artificial Microstructure and Mesoscopic Physics, School of Physics, Peking University, Beijing, China [5]Peking-Tsinghua Joint Center for Life Sciences, Peking University, Beijing, China [6]Pengcheng Laboratory, Shenzhen, China [7]School of Physics, Peking University, Beijing, China [8]Center for Quantitative Biology, Peking University, Beijing, China [9]National Biomedical Imaging Center, Peking University, Beijing, China [10]School of Intelligence Science and Engineering, Harbin Institute of Technology, Shenzhen, China. Correspondence to: Youdong Mao <ymao@pku.edu.cn>, Jie Chen <jiechen2019@pku.edu.cn>.

*Proceedings of the 43$^{rd}$ International Conference on Machine Learning*, Seoul, South Korea. PMLR 306, 2026. Copyright 2026 by the author(s).

pling Boltzmann distributions and generating kinetic trajectories. Early frameworks, such as Str2Str (Lu et al., 2024), utilize score-based matching to translate static structures into conformational ensembles. More recently, flow-matching models like AlphaFlow (Jing et al., 2024a) and P2DFlow (Jin et al., 2025) have integrated pre-trained structure prediction networks (Jumper et al., 2021) with SE(3)-equivariant flows to capture structural diversity. Despite their efficiency, these methods often inherit inductive biases from static training data (PDB) and struggle to maintain long-range kinetic coherence in complex temporal sequences. Furthermore, while recent trajectory generators like aSAMt (Janson et al., 2025b) and ConfRover (Shen et al., 2025) address temporal dynamics, they remain susceptible to error accumulation during long-horizon rollouts in the time domain.

A critical challenge in modeling protein trajectories is the multi-scale nature of their motion: proteins exhibit both high-frequency local fluctuations and low-frequency global collective motions(Cheatum, 2020). Existing time-domain autoregressive models often struggle to balance these disparate scales, leading to drift or loss of structural integrity over time. As illustrated in Figure 1, we posit that protein dynamics can be more effectively modeled in the frequency domain, where temporal signals are naturally disentangled into independent spectral components.

Beyond mitigating autoregressive drift, the frequency-domain view is also physically well aligned with *local near-equilibrium* protein dynamics (Ma, 2005; Bauer et al., 2019). After removing rigid-body motion, fluctuations around a local anchor conformation can be approximated as motion in a local harmonic well (Petrone & Pande, 2006; Ma, 2005), which is the basis of normal-mode theory(Hinsen, 2005). In this regime, spectral quantities are more directly tied to mode stiffness, damping, and frequency structure (Singh et al., 2018; Amer & Robicheaux, 2023), whereas in frame-wise coordinate prediction these factors remain entangled in the time-domain signal together with instantaneous phase and thermal fluctuations.

In this work, we propose **BioDynaSpec**, a novel generative framework that reformulates protein dynamics as a Spatio-Spectral Autoregressive Diffusion process. It is not merely a numerical reparameterization of protein dynamical trajectory generation; inspired by a physical perspective, BioDynaSpec is designed as an *anchor-conditioned local spectral dynamics* model. As illustrated in Figure 2, our approach introduces Independent Windowed Fourier Decomposition (IWFD) to partition long trajectories into windows, converting coordinate time-series into spectral tokens. This shift allows the model to perform frequency-wise generation, reconstructing structural details from low to high frequencies to ensure global coherence while capturing

fine-grained vibrations. To enforce physical consistency, we introduce Inter-Residue Frequency Coupling (IRFC), which injects a learnable, distance-aware Gaussian bias into the attention mechanism. Motivated by the harmonic approximation in molecular mechanics, IRFC adaptively modulates spatial couplings—applying strong constraints for local covalent/hydrogen bonds and broader kernels for distant non-bonded interactions.

Our main contributions are summarized as follows: (1) We introduce **Independent Windowed Fourier Decomposition (IWFD)**, which reformulates protein trajectory generation as a window-wise frequency-domain reconstruction problem to mitigate long-horizon error accumulation. (2) We provide a **local near-equilibrium spectral perspective** for protein dynamics modeling, where per-window aligned and anchor-conditioned spectra more directly encode local mode structure than frame-wise coordinate prediction. (3) We propose **Inter-Residue Frequency Coupling (IRFC)**, a physics-inspired attention bias with learnable Gaussian distance kernels for modeling multi-scale mechanical heterogeneity in protein structures. (4) We design a **Spatio-Spectral Hybrid Transformer** that factorizes spatial and spectral dependencies and combines low-to-high frequency autoregression with diffusion denoising for spectral token generation. (5) Extensive experiments on ATLAS show that BioDynaSpec outperforms strong flow-matching and diffusion baselines on both trajectory generation and equilibrium distributional accuracy.

## 2. Related Work

### 2.1. Protein Conformational Sampling

AlphaFold2 (Jumper et al., 2021) has largely solved single-structure prediction, but protein function is often determined by an ensemble of conformations (Kokkinidis et al., 2012). Recent work therefore targets sampling the Boltzmann ensemble with deep generative models.

**Inference-based sampling and flow matching.** Several methods adapt static predictors to generate diverse conformers. AlphaFlow and ESMFlow (Jing et al., 2024a) use flow matching to fine-tune AlphaFold (Jumper et al., 2021) and ESMFold (Lin et al., 2023), enabling faster and more diverse sampling than heuristic MSA subsampling. Protenix (Team et al., 2025) extends this line to biomolecular complexes with a diffusion transformer. These approaches typically deliver high structural fidelity but often rely on strong structural priors to preserve physical validity.

**Physically-consistent generation.** To better respect the underlying energy landscape, CONFDIFF (Wang et al., 2024) proposes a force-guided SE(3) diffusion method that reweights scores using physical force fields. BioEmu (Lewis et al., 2025) further connects generation to thermodynam-

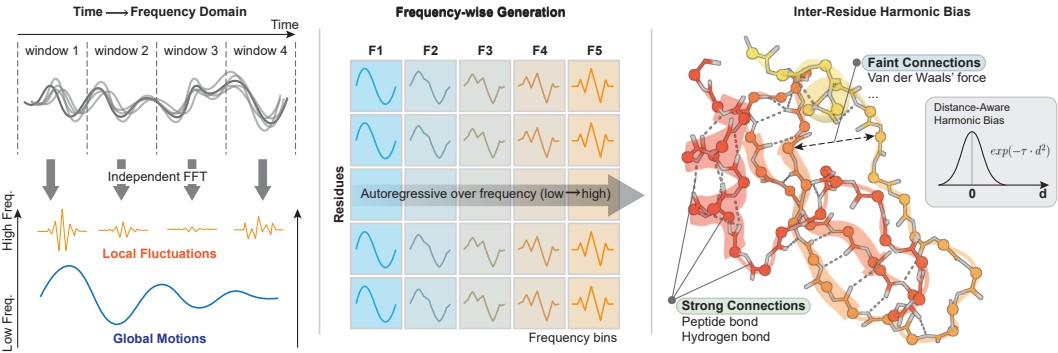

*Figure 1.* Motivation of BioDynaSpec. **(Left)** IWFD transforms windowed trajectories into the frequency domain to disentangle global motions from local fluctuations. **(Middle)** Spectral tokens are generated via a low-to-high frequency autoregressive schedule to preserve physical hierarchy. **(Right)** IRFC enforces geometric inductive biases using a distance-aware harmonic term, , which adaptively modulates spatial interactions based on physical connection strengths.

ics, predicting relative free energies at accuracy comparable to millisecond-scale MD. Environmental conditioning has also been explored: aSAMt (Janson et al., 2025a) is a transferable generator conditioned on temperature for probing thermal-dependent flexibility. However, most of these models emphasize i.i.d. sampling from equilibrium ensembles, while temporal transition paths (kinetics) remain less directly modeled.

### 2.2. Protein Dynamics Trajectory Generation

Trajectory generation is harder than i.i.d. sampling because it must satisfy per-frame geometric constraints while capturing temporal dependencies.

**Autoregressive and video-style models.** Inspired by video generation, MDGEN (Jing et al., 2024b) uses an SE(3)-equivariant diffusion model to sample trajectories conditioned on keyframes, with current emphasis on smaller systems such as tetrapeptides. ConfRover (Shen et al., 2025) combines a causal transformer for temporal modeling with a diffusion decoder for backbone generation (Shen et al., 2025), and supports variable-length trajectories; however, its discrete time steps can limit modeling at arbitrary temporal resolutions.

**Continuous-time and prior-guided flows.** P2DFlow (Jin et al., 2025) formulates generation via SE(3) flow matching by solving a continuous ODE, and introduces a mixed prior (structural prediction plus noise) to capture transient contacts and fluctuations observed in MD (Jin et al., 2025). More broadly, modeling the full spectral range of protein motion—from high-frequency bond vibrations to low-frequency domain rearrangements—remains challenging for standard transformers. Theory also indicates that RoPE (Su et al., 2024) can induce oscillatory attention under extreme extrapolation, whereas ALiBi (Press et al., 2022) yields a monotonic decay that better matches physical locality.

Overall, a remaining gap is frequency-domain trajectory modeling that decouples motion modes, beyond purely time-domain sequence learning. While EigenFold (Jing et al., 2023) and HarmonicFlow (Stark et al., 2024) use harmonic-oscillator priors for structure, they do not explicitly formulate dynamics in spectral space. BioDynaSpec fills this gap with a spatio-spectral transformer and harmonic priors to better learn multi-scale motions and improve extrapolation stability.

## 3. Methods

### 3.1. Problem Setup

We study long-horizon protein dynamics from a trajectory $\mathcal{T} = \{\mathbf{r}_t\}_{t=1}^T$, where $\mathbf{r}_t \in \mathbb{R}^{N \times 3}$ denotes the Cartesian coordinates of $N$ atoms at time $t$. BioDynaSpec models *residue-level* dynamics using C$\alpha$ coordinates. Let $N_r$ be the number of residues and $\mathbf{x}_t \in \mathbb{R}^{N_r \times 3}$ denote the C$\alpha$ coordinates at time $t$. Given sequence-level context $\mathbf{C}_{\text{seq}}$ and an initial conformation, our goal is to learn a conditional generative model that can roll out long-horizon dynamics. To mitigate error accumulation from time-domain autoregression, we operate on *independent windows* and generate each window in the *frequency domain*, while rolling out windows sequentially through a structural anchor. The complete training procedure is summarized in Algorithm 1, while the inference (rollout) procedure is detailed in Algorithm 2.

### 3.2. Physical Motivation: Local Near-Equilibrium Spectral Representation

BioDynaSpec is designed to model *local near-equilibrium dynamics* within each aligned temporal window. Let $\mathbf{x}_{\text{start}}^{(k)} \in \mathbb{R}^{N_r \times 3}$ denote the anchor conformation of window $k$, and let $\delta\mathbf{x}_t^{(k)} = \tilde{\mathbf{X}}_{t,:,:}^{(k)} - \mathbf{x}_{\text{start}}^{(k)}$ denote the fluctuation around this anchor.

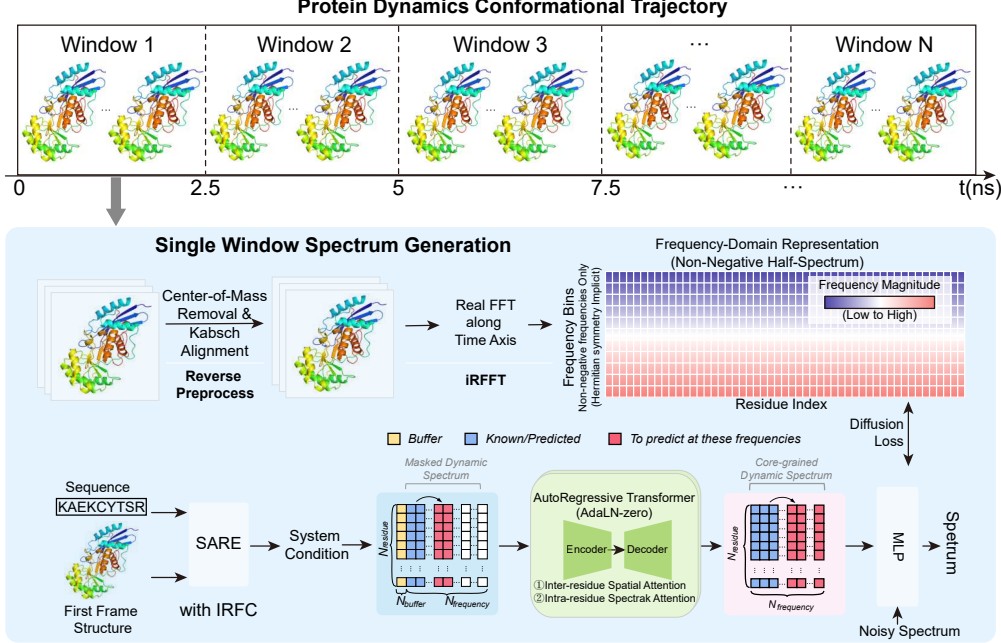

*Figure 2.* Overview of the BioDynaSpec framework. **(Top)** Partition trajectories into independent temporal windows. **(Middle)** Align each window (COM removal + Kabsch) and apply time-axis RFFT to obtain a non-negative half-spectrum. **(Bottom)** Pipeline: (i) SARE+IRFC conditioner for sequence/initial-structure priors; (ii) spatio-spectral autoregressive transformer (AdaLN-zero) for inter-residue spatial and intra-residue spectral dependencies; (iii) diffusion MLP denoiser generating spectral tokens from low to high frequencies.

Under a local near-equilibrium assumption, the potential energy around the anchor can be approximated by a quadratic form

$$U\left(\mathbf{x}_{\text{start}}^{(k)} + \delta\mathbf{x}\right) \approx U\left(\mathbf{x}_{\text{start}}^{(k)}\right) + \frac{1}{2}\,\delta\mathbf{x}^\top \mathbf{H}^{(k)}\delta\mathbf{x}, \quad (1)$$

where $\mathbf{H}^{(k)}$ is the local Hessian of the potential energy surface at the anchor conformation.

The resulting local dynamics can be described through a Langevin approximation,

$$\mathbf{M}\,\ddot{\delta\mathbf{x}} + \mathbf{\Gamma}\,\dot{\delta\mathbf{x}} + \mathbf{H}^{(k)}\delta\mathbf{x} = \boldsymbol{\eta}(t), \quad (2)$$

where $\mathbf{M}$ is the mass matrix, $\mathbf{\Gamma}$ is a damping operator, and $\boldsymbol{\eta}(t)$ is thermal noise. After projection onto a local mode $q_\ell$, the corresponding power spectral density takes the form

$$S_\ell(\omega) \propto \frac{2k_B T \gamma_\ell}{m_\ell^2(\omega_\ell^2 - \omega^2)^2 + \gamma_\ell^2\omega^2}, \quad (3)$$

showing that the spectrum directly reflects local stiffness, damping, and mode structure. This motivates a spectral view of local protein fluctuations. Within a finite window, the spectrum makes near-equilibrium mode structure more explicit, whereas frame-wise coordinates leave it entangled with instantaneous phase and thermal noise in the time domain. Accordingly, BioDynaSpec models protein dynamics

as a sequence of anchor-centered local fluctuation processes in spectral space, while the concrete window decomposition and conditional rollout are introduced in the following subsections.

### 3.3. Independent Windowed Fourier Decomposition (IWFD)

**Per-window preprocessing.** For each residue $i \in \{1, \ldots, N_r\}$, we extract its C$\alpha$ coordinate from the all-atom trajectory:

$$\mathbf{x}_{t,i} = \mathbf{r}_{t,i}^{(C\alpha)} \in \mathbb{R}^3, \qquad \mathbf{x}_t = [\mathbf{x}_{t,1}; \ldots; \mathbf{x}_{t,N_r}] \in \mathbb{R}^{N_r \times 3}, \quad (4)$$

and denote the residue-level trajectory by $\mathcal{X} = \{\mathbf{x}_t\}_{t=1}^T$. We divide $\mathcal{X}$ into $M$ non-overlapping windows of length $W$ and assume $T = MW$. For window $k \in \{1, \ldots, M\}$, define the windowed coordinates $\mathbf{X}^{(k)} \in \mathbb{R}^{W \times N_r \times 3}$ by

$$\mathbf{X}_{t,i,:}^{(k)} \triangleq \mathbf{x}_{(k-1)W+t,\,i} \in \mathbb{R}^3, \qquad t \in \{1, \ldots, W\}, \quad (5)$$

where $\mathbf{X}_{t,i,c}^{(k)}$ denotes axis $c \in \{x, y, z\}$. Within each window, we remove the center of mass and apply Kabsch alignment to the first frame of the same window, yielding the aligned trajectory $\tilde{\mathbf{X}}^{(k)} \in \mathbb{R}^{W \times N_r \times 3}$. This suppresses global rigid-body motion, ensuring that the resulting frequency spectrum reflects intrinsic fluctuations.

**Per-window Real FFT (rFFT) and inverse real FFT (irFFT).** For each window $k$, residue $i$, and axis $c \in \{x, y, z\}$, let $\tilde{\mathbf{X}}^{(k)}_{\cdot,i,c} \in \mathbb{R}^W$ be the length-$W$ real-valued time series. Let $\Phi$ denote the real-input FFT (rFFT) along the time dimension, and let $F = \lfloor W/2 \rfloor + 1$ be the number of non-negative frequency bins. We compute the rFFT coefficients $\mathbf{S}^{(k)} = \Phi(\tilde{\mathbf{X}}^{(k)}) \in \mathbb{C}^{F \times N_r \times 3}$, whose entries for $f \in \{0, \ldots, F-1\}$ are

$$\mathbf{S}^{(k)}_{f,i,c} = \sum_{t=1}^{W} \tilde{\mathbf{X}}^{(k)}_{t,i,c} \exp\left(-i\frac{2\pi}{W}(t-1)f\right). \qquad (6)$$

Since $\tilde{\mathbf{X}}^{(k)}_{\cdot,i,c}$ is real-valued, the full DFT satisfies conjugate symmetry and the time-domain signal is reconstructed by the inverse real FFT ($\Phi^{-1}$):

$$\tilde{\mathbf{X}}^{(k)}_{\cdot,i,c} = \Phi^{-1}\left(\mathbf{S}^{(k)}_{\cdot,i,c}; W\right) \in \mathbb{R}^W. \qquad (7)$$

**Per-frequency normalization.** We normalize rFFT coefficients per frequency to keep a stable scale across bins. For each frequency bin $f \in \{0, \ldots, F-1\}$, we compute a training-set scaling factor from the magnitudes of all coefficients at that frequency:

$$m^{(k)}_{f,i,c} \triangleq \left|\mathbf{S}^{(k)}_{f,i,c}\right| = \sqrt{\Re\left(\mathbf{S}^{(k)}_{f,i,c}\right)^2 + \Im\left(\mathbf{S}^{(k)}_{f,i,c}\right)^2}, c \in \{x, y, z\}. \qquad (8)$$

Let $s_f$ be the 95-th percentile of $\{m^{(k)}_{f,i,c}\}$ over all training windows $k$ and all residues $i$ and axes $c$. We then normalize

$$\tilde{\mathbf{S}}^{(k)}_{f,i,c} \triangleq \frac{\mathbf{S}^{(k)}_{f,i,c}}{s_f}. \qquad (9)$$

During inference, we undo this scaling by $\mathbf{S}^{(k)}_{f,i,c} = s_f \tilde{\mathbf{S}}^{(k)}_{f,i,c}$ before applying the inverse rFFT.

**Real-valued tokenization.** We convert each normalized complex coefficient $\tilde{\mathbf{S}}^{(k)}_{f,i,c}$ into a real-valued token by concatenating its real and imaginary parts. For each $(i, f)$,

$$\mathbf{X}^{(k)}_{\text{tok}}[i, f, :] = \left[\Re\left(\tilde{\mathbf{S}}^{(k)}_{f,i,x}\right), \Re\left(\tilde{\mathbf{S}}^{(k)}_{f,i,y}\right), \Re\left(\tilde{\mathbf{S}}^{(k)}_{f,i,z}\right),\right.$$
$$\left.\Im\left(\tilde{\mathbf{S}}^{(k)}_{f,i,x}\right), \Im\left(\tilde{\mathbf{S}}^{(k)}_{f,i,y}\right), \Im\left(\tilde{\mathbf{S}}^{(k)}_{f,i,z}\right)\right] \in \mathbb{R}^6. \qquad (10)$$

This yields a token tensor

$$\mathbf{X}^{(k)}_{\text{tok}} \in \mathbb{R}^{N_r \times F \times d_t}, \qquad d_t = 6. \qquad (11)$$

**Independent windows.** We model and generate each non-overlapping window independently. We do not use overlap-add STFT reconstruction, because inconsistent predicted phases in overlapped regions can lead to geometric inconsistencies after reconstruction. We evaluate this design choice in Appendix C.

## 3.4. Conditional Generation: Window-wise Dynamics Spectrum Prediction

For each time-domain window $k$, our generation model predicts the normalized rFFT coefficients $\tilde{\mathbf{S}}^{(k)} \in \mathbb{C}^{F \times N_r \times 3}$ conditioned on the protein sequence representation $\mathbf{C}_{\text{seq}}$ and an anchor structure. Let $\mathbf{x}^{(k)}_{\text{start}} \in \mathbb{R}^{N_r \times 3}$ denote the anchor frame for window $k$, defined as the first aligned $C\alpha$ frame of that window, $\mathbf{x}^{(k)}_{\text{start}} \triangleq \tilde{\mathbf{X}}^{(k)}_{1,:,:}$. We factorize the conditional distribution over window-wise normalized spectra as

$$p\left(\{\tilde{\mathbf{S}}^{(k)}\}_{k=1}^M \mid \mathbf{C}_{\text{seq}}\right) = \prod_{k=1}^{M} p_\theta\left(\tilde{\mathbf{S}}^{(k)} \mid \mathbf{x}^{(k)}_{\text{start}}, \mathbf{C}_{\text{seq}}\right). \qquad (12)$$

At rollout time, we first undo the per-frequency normalization and then apply the inverse rFFT. For each $k$ and $f \in \{0, \ldots, F-1\}$,

$$\hat{\mathbf{S}}^{(k)}_{f,i,c} = s_f \hat{\tilde{\mathbf{S}}}^{(k)}_{f,i,c}, \qquad i \in \{1, \ldots, N_r\}, \; c \in \{x, y, z\}, \qquad (13)$$

and decode to the time-domain window

$$\hat{\mathbf{X}}^{(k)} = \Phi^{-1}\left(\hat{\mathbf{S}}^{(k)}; W\right) \in \mathbb{R}^{W \times N_r \times 3}. \qquad (14)$$

We then set the next anchor recursively using the last frame of the generated window:

$$\mathbf{x}^{(k+1)}_{\text{start}} = \hat{\mathbf{X}}^{(k)}_{W,:,:} \in \mathbb{R}^{N_r \times 3}. \qquad (15)$$

## 3.5. Structure-Aware Representation Encoder (SARE)

To encode geometric and evolutionary constraints, we employ the Structure-Aware Representation Encoder (SARE). As detailed in Appendix A, SARE maps the anchor conformation $\mathbf{x}^{(k)}_{\text{start}}$ and sequence context $\mathbf{C}_{\text{seq}}$ (derived from ESM-2) into a conditioning representation $\mathbf{a}^{(k)}$. The encoding consists of three stages: (i) multimodal feature initialization refined by triangular self-attention; (ii) Invariant Point Attention (IPA) for extracting SE(3)-invariant local geometric features; and (iii) a Spatial Transformer for global feature aggregation. Within the Spatial Transformer stage, we introduce the Inter-Residue Frequency Coupling (IRFC) module to modulate attention based on spatial proximity.

## 3.6. Inter-Residue Frequency Coupling (IRFC)

To operationalize the local near-equilibrium view in the network architecture, we introduce Inter-Residue Frequency Coupling (IRFC), a structure-aware attention bias motivated by the local harmonic approximation in Sec. 3.2. Intuitively, near a stable conformation, coherent fluctuations are shaped by local stiffness and geometric coupling: stiff local interactions mainly support high-frequency vibrations, while softer

long-range couplings govern lower-frequency collective motions (McCammon et al., 1977; Fenwick, 2021). IRFC encodes this intuition through a learnable distance-aware Gaussian penalty in attention logits.

**Gaussian-biased attention.** Let $\mathbf{r}_i \triangleq \mathbf{x}_{\text{start}}^{(k)}[i, :] \in \mathbb{R}^3$ be the C$\alpha$ coordinate of residue $i$ in the anchor frame of window $k$, and let $d_{ij} = \|\mathbf{r}_i - \mathbf{r}_j\|_2$. For attention head $h$, we compute

$$\alpha_{ij}^{(h)} = \text{softmax}_j \left( \frac{\mathbf{q}_i^{(h)\top} \mathbf{k}_j^{(h)}}{\sqrt{d}} + \mathbf{b}_{ij}^{\text{pair}} - \tau_h d_{ij}^2 \right), \quad (16)$$
$$\tau_h = \text{softplus}(\gamma_h),$$

where $\tau_h \geq 0$ is a head-specific stiffness parameter derived from a learnable scalar $\gamma_h$. We initialize $\gamma_h = -2.0$ to start with a broad receptive field.

**Physical Interpretation.** A large $\tau_h$ yields a narrow Gaussian kernel and enforces local attention (stiff constraints), while a small $\tau_h$ allows long-range attention (global collective modes). Multiple heads therefore implement multiscale geometric coupling.

**Design Choice: Gaussian vs. Laplacian.** We use the squared-distance Gaussian form $\exp(-\tau d^2)$ (rather than $\exp(-\tau d)$) to match the quadratic structure implied by the harmonic approximation, while providing smooth gradients and suppressing irrelevant long-range interactions that contradict the anchor geometry.

### 3.7. Spatial-Spectral Autoregressive Diffusion Modeling

Given the conditioning representation $\mathbf{a}^{(k)}$, we generate real-valued spectral tokens $\hat{\mathbf{X}}_{\text{tok}}^{(k)} \in \mathbb{R}^{N_r \times F \times 6}$ (equivalently, $\hat{\tilde{\mathbf{S}}}^{(k)}$) progressively from low to high frequencies.

**Progressive Masking with Self-Forcing.** We use a two-phase training strategy.

In Phase I (Progressive Masking), we randomly sample a frequency cutoff $s \in \{0, \dots, F\}$ and retain the first $s$ low-frequency tokens. Let

$$\mathbf{X}_{1:s}^{(k)} = \mathbf{X}_{\text{tok}}^{(k)}[:, 1\!:\!s, :] \in \mathbb{R}^{N_r \times s \times d_t}, \qquad d_t = 6. \quad (17)$$

We then construct a full-length coarse proxy by applying 1D linear interpolation along the frequency axis:

$$\tilde{\mathbf{X}}^{(k,s)} = \text{Interp}_{\text{freq}}\big(\mathbf{X}_{1:s}^{(k)}, F\big) \in \mathbb{R}^{N_r \times F \times d_t}. \quad (18)$$

Here, $\text{Interp}_{\text{freq}}(\cdot, F)$ denotes linear interpolation along the frequency dimension, applied independently for each residue and token channel. The resulting $\tilde{\mathbf{X}}^{(k,s)}$ is used

only as a *coarse conditioning signal* that provides a smooth low-resolution spectral scaffold for the planner in Phase I. Importantly, this interpolation step is *not* intended to reconstruct the missing high-frequency coefficients: the supervision target remains the full ground-truth spectrum, and the unresolved high-frequency bands are still predicted by the model.

To reduce the train–test mismatch between random masking in Phase I and sequential low-to-high inference at test time, we adopt Phase II (Self-Forcing) (Huang et al., 2025), where the model follows the autoregressive inference schedule by predicting the next frequency bin conditioned on the ground-truth history.

**Spatio-spectral Hybrid Planner.** The planner $\mathcal{P}_\theta$ follows an encoder–decoder design. The encoder integrates masked spectral tokens with the structural condition $\mathbf{a}^{(k)}$, and the decoder autoregressively produces conditioning features $\mathbf{c}^{(k)}$. We employ Hybrid Spatio-Spectral Blocks that alternate between: (1) **Inter-residue spatial attention** across residues ($N_r$) to capture collective motions; and (2) **Intra-residue spectral attention** along the frequency axis ($F$) with Rotary Position Embeddings (RoPE) to model phase correlations within each residue.

**Diffusion Head for Continuous Tokens.** Instead of directly regressing continuous tokens with MSE, we use a diffusion head conditioned on $\mathbf{c}$ (Figure 2, bottom right). Let $\mathbf{x}_0$ denote the clean spectral token and $\mathbf{x}_\sigma = \mathbf{x}_0 + \sigma \boldsymbol{\epsilon}$ a noisy version with $\boldsymbol{\epsilon} \sim \mathcal{N}(\mathbf{0}, \mathbf{I})$. The denoising network $F_\theta$ is trained with

$$\mathcal{L}_{\text{diff}} = \mathbb{E}_{\sigma, \boldsymbol{\epsilon}} \left[ \lambda(\sigma) \left\| F_\theta(\mathbf{x}_\sigma; \sigma, \mathbf{c}) - \mathbf{x}_0 \right\|_2^2 \right]. \quad (19)$$

During inference, we sample tokens using a 50-step scheduler and linearly interpolate between frequency levels to obtain a smooth low-to-high generation trajectory.

### 3.8. Real-sequence Reconstruction via iRFFT ($\Phi^{-1}$)

Given generated tokens $\hat{\mathbf{X}}_{\text{tok}}^{(k)} \in \mathbb{R}^{N_r \times F \times 6}$, we first recover the normalized complex half-spectrum $\hat{\tilde{\mathbf{S}}}^{(k)} \in \mathbb{C}^{F \times N_r \times 3}$ by inverting the real/imaginary concatenation (Eq. 10). For axis $x$,

$$\hat{\tilde{\mathbf{S}}}_{f,i,x}^{(k)} = \hat{\mathbf{X}}_{\text{tok}}^{(k)}[i, f, 1] + i\, \hat{\mathbf{X}}_{\text{tok}}^{(k)}[i, f, 4], \quad (20)$$

and similarly for $(y, z)$ using indices $(2, 5)$ and $(3, 6)$.

We then undo the per-frequency normalization using the scaling factors $\{s_f\}_{f=0}^{F-1}$:

$$\hat{\mathbf{S}}_{f,i,c}^{(k)} = s_f\, \hat{\tilde{\mathbf{S}}}_{f,i,c}^{(k)}, \qquad c \in \{x, y, z\}. \quad (21)$$

The inverse real FFT reconstructs a length-$W$ real sequence:

$$\hat{\mathbf{X}}^{(k)} = \Phi^{-1}\left(\hat{\mathbf{S}}^{(k)}; W\right) \in \mathbb{R}^{W \times N_r \times 3}. \quad (22)$$

**Inverting the Per-window Rigid Transform.** During preprocessing of window $k$, we store the rigid transform $(\mathbf{R}^{(k)}, \mathbf{t}^{(k)})$ applied to map original coordinates to the aligned frame, so that

$$\tilde{\mathbf{X}}_{t,i,:}^{(k)} = \mathbf{R}^{(k)}\mathbf{X}_{t,i,:}^{(k)} + \mathbf{t}^{(k)}, \qquad \mathbf{R}^{(k)} \in SO(3),\ \mathbf{t}^{(k)} \in \mathbb{R}^3. \quad (23)$$

After generation, we map the reconstructed window back to the original coordinate frame by

$$\mathbf{X}_{t,i,:}^{(k)} = \mathbf{R}^{(k)\top}\left(\hat{\mathbf{X}}_{t,i,:}^{(k)} - \mathbf{t}^{(k)}\right). \quad (24)$$

We then update the next anchor as in Eq. 15.

## 4. Experiment

### 4.1. Experiment Setups

#### 4.1.1. DATASETS

We train and evaluate our method primarily on the ATLAS dataset (Vander Meersche et al., 2024), a large-scale collection of 100 ns molecular dynamics trajectories. We choose ATLAS for its diversity, containing proteins with lengths up to 1000 residues (we exclude 4 sequences exceeding 1000 residues), which allows us to test the generalizability of our model across a wide range of protein sizes. The dataset comprises 1,390 trajectories, which we split into training (1,269 trajectories), validation (39), and test (82) sets. Each trajectory provides sequential 3D structures treated as ground truth.

#### 4.1.2. IMPLEMENTATION DETAILS

**Data Preprocessing.** We preprocess the raw trajectories using the Independent Windowed Fourier Decomposition (IWFD) method (see subsection 3.3 for details). For all experiments on ATLAS, we employ a fixed window size of $W = 2500$ frames with a hop ratio of 1.0 (i.e., no overlap). The time step is set to $\Delta t = 10\,\text{ps}$, matching the original MD simulation configuration. The normalization scaling factors $s_f$ for each frequency are computed solely using the training set. For structure-aware representations, we utilize ESM-2-650M (Lin et al., 2023) to extract initial residue embeddings.

**Model Architecture.** We implement the model architecture described in Sections 3.4–3.6[1]. All transformer modules use

---

[1]See also: subsection 3.5 (Structure-Aware Representation Encoder (SARE) ($\mathcal{E}$)), subsection 3.6 (Inter-Residue Frequency Coupling (IRFC): Harmonic-informed Distance Bias for Structure-Aware Attention), and subsection 3.7 (Spatial-Spectral Autoregressive Diffusion Modeling) for additional details.

a hidden dimension of $d_{\text{model}} = 768$ with 12 attention heads. The decoder's diffusion head operates over complex-valued spectral tokens (dimension $d_t = 6$). We employ gradient checkpointing and chunked processing (size 2560) for memory efficiency. The diffusion process uses $T = 1000$ steps with a cosine noise schedule (Ho et al., 2020) during training, and 50 sampling steps via Denoising Diffusion Implicit Models (DDIM) (Song et al., 2022) at inference.

**Optimization.** Models are trained using PyTorch Lightning (Paszke et al., 2019) with the AdamW optimizer (Loshchilov & Hutter, 2019) (learning rate $1 \times 10^{-4}$, betas (0.9, 0.999)). We use a warmup-cosine learning rate schedule with plateau reduction (warmup for 10 epochs from $1 \times 10^{-7}$), combined with a ReduceLROnPlateau scheduler (Al-Kababji et al., 2022) (patience 8, factor 0.7). Gradient clipping is applied at value 1.0 to stabilize training. We train for a maximum of 50 epochs with batch size 1 (due to variable sequence lengths), using mixed precision training (bfloat16) to accelerate computation. We use 32 data loading workers and set the random seed to 42 for reproducibility, enabling deterministic operations where possible.

**Computational Setup.** All experiments were conducted on a node with $8\times$ NVIDIA H200 GPUs (143 GB each). We use PyTorch 2.6.0, CUDA 12.4, and Python 3.10.19. Training typically required 2–5 days per model.

#### 4.1.3. BASELINES

For trajectory generation, we compare against ProAR (Cheng et al., 2026) and MDGEN (Jing et al., 2024b). MDGEN (Jing et al., 2024b) is a flow-based generative model trained on the ATLAS dataset. ProAR (Cheng et al., 2026) is a probabilistic autoregressive model for protein dynamics trajectory generation on ATLAS, using Gaussian frame modeling with anti-drifting sampling. For conformational sampling, we compare with two specialized samplers: AlphaFlow (Jing et al., 2024a) (flow-based) and CONFDIFF (Wang et al., 2024) (diffusion-based), both of which are time-independent and fine-tuned on ATLAS. We also include MDGEN (Jing et al., 2024b) and ProAR (Cheng et al., 2026) in this evaluation, as both support conformation sampling alongside trajectory generation. All baseline methods are evaluated using their publicly available implementations with default hyperparameters unless otherwise specified.

### 4.2. Main Results

We evaluate our model on two tasks for assessing protein dynamics modeling: (1) **trajectory generation**, which examines the model's ability to maintain dynamical consistency and explore conformational space as it evolves over time; and (2) **zero-shot conformational sampling**, which evaluates the diversity and physical realism of the generated

*Table 1.* Comparison of frame-wise C$\alpha$-RMSE between BioDynaSpec and baseline models, measured in angstroms (Å).

|  | $R_{50}$ | $R_{100}$ | $R_{150}$ | $R_{200}$ | $R_{250}$ |
|---|---|---|---|---|---|
| MDGEN | 3.184 | 3.461 | 3.607 | 3.735 | 3.813 |
| ProAR | 3.120 | 3.292 | 3.380 | 3.472 | 3.529 |
| BioDynaSpec | **1.256** | **1.371** | **1.431** | **1.482** | **1.509** |

conformational ensemble against state-of-the-art baseline methods.

### 4.2.1. TRAJECTORY GENERATION

**Trajectory Reconstruction Fidelity.** For each test protein, we generate a trajectory of 250 frames at 400 ps per frame from the initial structure of each of its three MD replicates, matching the 100 ns ATLAS reference trajectories. We adopt this setting because MDGEN, one of our baselines, only supports trajectories of length 250 with 400 ps spacing. To quantify reconstruction fidelity, we report the average frame-wise C$\alpha$-RMSE over the first $s$ frames, denoted by $R_s$.

For each frame $i$, we first align the generated structure to the reference on C$\alpha$ atoms using the Kabsch algorithm, and then compute

$$\text{RMSE}_i = \sqrt{\frac{1}{N_\alpha} \sum_{j=1}^{N_\alpha} \left\| \mathbf{x}_{i,j}^{\text{gen}} - \mathbf{x}_{i,j}^{\text{ref}} \right\|_2^2}, \qquad (25)$$

where $N_\alpha$ is the number of C$\alpha$ atoms and $\mathbf{x}_{i,j}^{\text{gen}}, \mathbf{x}_{i,j}^{\text{ref}} \in \mathbb{R}^3$ are the aligned coordinates of atom $j$ at frame $i$. We then define

$$R_s = \frac{1}{s} \sum_{i=1}^{s} \text{RMSE}_i. \qquad (26)$$

As shown in Table 1, BioDynaSpec achieves substantially lower errors than MDGEN and ProAR across all horizons. In particular, at $s = 250$ BioDynaSpec reaches $R_{250} = 1.509$ Å, reducing error by 60.4% relative to MDGEN and 57.2% relative to ProAR. These results indicate that BioDynaSpec maintains high trajectory fidelity not only in the short term (e.g., $R_{50}$) but also over long rollouts (up to $R_{250}$), supporting the effectiveness of our frequency-domain autoregressive–diffusion generation with explicit structure conditioning.

**Generated Motion Pattern Accuracy** Because both MD and our model generate stochastic trajectories, frame-wise RMSE alone cannot measure whether a method matches the *distribution* of motions. We therefore compare generated and reference trajectories in two PCA-defined 2D spaces constructed from the three reference MD replicates for each protein: one based on C$\alpha$ Cartesian coordinates

*Table 2.* Pearson correlation of displacement profiles between generated and reference trajectories in PCA-2D space. Higher is better.

|  | PCA-2D(C$\alpha$ pwd.) | | PCA-2D(C$\alpha$ coord.) | |
|---|---|---|---|---|
|  | Ref. ($\uparrow$) | Step. ($\uparrow$) | Ref. ($\uparrow$) | Step. ($\uparrow$) |
| MDGEN | 0.105 | 0.024 | 0.115 | 0.014 |
| ProAR | 0.085 | -0.004 | 0.076 | -0.003 |
| BioDynaSpec | **0.170** | **0.456** | **0.181** | **0.452** |

*Table 3.* Hausdorff distance between generated and reference trajectories in PCA-2D space. Lower is better.

|  | PCA-2D(C$\alpha$ pwd.) | | PCA-2D(C$\alpha$ coord.) | |
|---|---|---|---|---|
|  | Ref. ($\downarrow$) | Step. ($\downarrow$) | Ref. ($\downarrow$) | Step. ($\downarrow$) |
| MDGEN | 17.071 | 6.581 | 1.576 | 0.721 |
| ProAR | 9.574 | 8.032 | 1.014 | 1.048 |
| BioDynaSpec | 24.697 | **4.724** | 2.505 | **0.487** |

and the other on C$\alpha$ pairwise distances (pwd.). Generated trajectories are projected into the same spaces.

To capture different temporal scales, we compute two displacement signals for each trajectory: (1) *reference displacement* (Ref.), the $\ell_2$ distance from each frame to the initial frame, reflecting long-range conformational exploration; and (2) *stepwise displacement* (Step.), the $\ell_2$ distance between consecutive frames, reflecting local dynamical consistency. We quantify distributional mismatch by the Hausdorff distance between generated and reference point sets in the corresponding PCA-2D space (lower is better), and temporal agreement by the Pearson correlation between generated and reference displacement sequences. All metrics are computed on 250-frame reference trajectories and averaged over test proteins.

As shown in Table 2, BioDynaSpec achieves the highest Reference and Stepwise Pearson correlations under both PCA representations (C$\alpha$ coord. and C$\alpha$ pwd.), with especially large gains on Stepwise correlation, indicating better temporal coupling of local dynamics than MDGEN and ProAR. Table 3 further shows that BioDynaSpec attains the lowest Stepwise Hausdorff distance under both representations, demonstrating that its local displacement distribution is closer to true MD trajectories. Although there remains room to improve alignment in long-horizon reference displacement distributions, BioDynaSpec consistently provides the most accurate and time-coherent local dynamics across both PCA representations.

### 4.2.2. CONFORMATIONAL SAMPLING

*Table 4.* Results on the ATLAS test set. Performance comparison of methods for modeling protein dynamics across different metrics.

| Metrics | AlphaFlow | ConfDiff | ConfRover | MDGen | BioDynaSpec |
|---|---|---|---|---|---|
| Root mean $W_2$-dist. $\downarrow$ | 2.64 | 2.75 | 2.62 | 2.81 | **1.31** |
| MD PCA $W_2$-dist. $\downarrow$ | 1.52 | 1.41 | 1.39 | 1.95 | **0.90** |
| Joint PCA $W_2$-dist. $\downarrow$ | 2.29 | 2.27 | 2.28 | 2.38 | **1.19** |

We evaluate equilibrium conformational sampling on the ATLAS test set, following the AlphaFlow benchmark protocol (Jing et al., 2024a). For each protein, AlphaFlow, ConfDiff, and ConfRover generate 250 i.i.d. samples. MDGen and BioDynaSpec generate a 250-frame trajectory initialized from the first MD frame. Table 4 reports distributional metrics comparing generated ensembles to the reference MD.

BioDynaSpec matches or exceeds specialized sampling baselines on all three metrics, achieving the best Root Mean $W_2$ (1.31), MD PCA $W_2$ (0.90), and Joint PCA $W_2$ (1.19). Compared with the next best method, this corresponds to relative reductions of $50.03\%$, $35.25\%$, and $47.58\%$, respectively. Thus, even under trajectory-based generation from a fixed initial conformation, BioDynaSpec produces ensembles that are closer to the reference equilibrium distribution.

### 4.2.3. NEAR-EQUILIBRIUM LOCAL DYNAMICS AND COVARIANCE CONSISTENCY

To further test whether the generated trajectories capture the structure of *local near-equilibrium fluctuations*, we introduce two groups of extended metrics.

**Local dynamics fidelity.** We project both generated and reference trajectories onto the ground-truth (GT) PCA modes and GT tICA modes, and compare their mode-resolved power spectral densities (PSDs). Specifically, we report PCA-PSD-L1 and tICA-PSD-L1, the average $\ell_1$ distance between generated and reference PSDs in the corresponding GT mode spaces, as well as PCA-PSD-LogCorr and tICA-PSD-LogCorr, the correlation between log power spectra.

**Covariance consistency.** We further evaluate whether the generated trajectories match the covariance geometry of the reference MD ensemble in the GT PCA subspace. For this, we report CFRE, the relative Frobenius error between covariance matrices, and Covariance-Eig-RRMSE, the relative RMSE between covariance eigenvalue spectra.

Table 5 summarizes the results. BioDynaSpec reaches PCA-PSD-LogCorr $= 0.817$ and CFRE $= 0.989$, corresponding to a $21.9\%$ gain and a $36.7\%$ reduction over the next best method, respectively, showing that it more accurately captures the spectral structure of near-equilibrium fluctuations and better matches the covariance geometry of the reference MD ensemble.

## 5. Conclusion

We proposed **BioDynaSpec**, a spatio-spectral generative framework for protein dynamics that combines **Independent Windowed Fourier Decomposition (IWFD)** with a frequency-domain autoregressive–diffusion generator, and incorporates **Inter-Residue Frequency Coupling (IRFC)**

*Table 5.* Local dynamics fidelity and covariance consistency of generated trajectories. For PSD-L1, CFRE, and Covariance-Eig-RRMSE, lower is better; for PSD-LogCorr, higher is better.

| Category | Metric | MDGEN | ProAR | BioDynaSpec |
|---|---|---|---|---|
| **Local Dynamics Fidelity** | PCA-PSD-L1 ↓ | 1.331 | 1.138 | **1.053** |
| | PCA-PSD-LogCorr ↑ | 0.395 | 0.670 | **0.817** |
| | tICA-PSD-L1 ↓ | 1.501 | 1.375 | **1.275** |
| | tICA-PSD-LogCorr ↑ | 0.380 | **0.662** | 0.661 |
| **Covariance Consistency** | CFRE ↓ | 1.563 | 2.030 | **0.989** |
| | Covariance-Eig-RRMSE ↓ | 1.184 | 1.625 | **0.941** |

as a structure-aware harmonic distance bias in attention. Beyond mitigating long-horizon drift, BioDynaSpec is motivated by a local near-equilibrium view of protein dynamics, where anchor-centered fluctuations are more naturally modeled in spectral space.

On **trajectory generation** over the ATLAS test set (82 proteins; 250 frames at 400 ps per frame), BioDynaSpec improves reconstruction fidelity across all horizons, reaching $R_{250} = 1.509\,\text{Å}$ and reducing error by $60.4\%$ relative to MDGEN and $57.2\%$ relative to ProAR. Beyond frame-wise RMSE, it also better captures *local temporal motion*, achieving the highest PCA-2D displacement-profile correlations and the lowest Stepwise Hausdorff distances, indicating more accurate short-range dynamical coupling.

For **equilibrium conformational sampling**, BioDynaSpec achieves the best distributional matching on the core $W_2$ metrics, with Root Mean $W_2 = 1.31$, MD PCA $W_2 = 0.90$, and Joint PCA $W_2 = 1.19$, corresponding to $50.03\%$, $35.25\%$, and $47.58\%$ improvements over the next best method, respectively. It also improves **near-equilibrium local dynamics and covariance consistency**, reaching PCA-PSD-LogCorr $= 0.817$ and CFRE $= 0.989$, indicating better mode-resolved spectral fidelity and closer covariance geometry matching to the reference MD ensemble.

Overall, these results support spatio-spectral modeling as an effective route to accurate and time-coherent protein dynamics generation, with improved local fluctuation fidelity and equilibrium distribution matching.

## 6. Limitations

BioDynaSpec is evaluated mainly on ATLAS, so generalization to substantially different environments still requires further validation. It also relies on a fixed-window, local near-equilibrium spectral approximation, which may be less accurate for very long-timescale transitions spanning multiple windows. Moreover, while recursive anchor updates maintain positional continuity, velocity and acceleration continuity are not explicitly enforced across window boundaries.

## Acknowledgements

This work was supported in part by National Key Research and Development Program of China (2023YFF1204400 and 2023YFF1204401), National Natural Science Foundation of China (12125401), the New Generation Artificial Intelligence-National Science and Technology Major Project (No. 2025ZD0122702), the Shenzhen Medical Research Funds in China (No. B2302037), Natural Science Foundation of China (No. 61972217, 32071459, 62176249, 62006133, 62271465), and AI for Science (AI4S)-Preferred Program, Peking University Shenzhen Graduate School, China, and the Shenzhen Loop Area Institute under grant FPF10120250014.

## Impact Statement

This paper presents work whose goal is to advance the field of Machine Learning. There are many potential societal consequences of our work, none which we feel must be specifically highlighted here.

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

## A. Structure-Aware Representation Encoder (SARE) ($\mathcal{E}$)

As shown in figure 3, the generation process is conditioned on a structure-aware latent representation $\mathcal{H}^{(k)}$ that encapsulates both evolutionary and geometric constraints. The SARE module $\mathcal{E}$ maps the starting anchor conformation $\mathbf{r}_{\text{start}}^{(k)}$ and the sequence context $\mathbf{C}_{\text{seq}}$ into this high-dimensional conditioning space:

$$\mathbf{a}^{(k)} = \mathcal{E}\big(\mathbf{r}_{\text{start}}^{(k)}, \mathbf{C}_{\text{seq}}\big), \qquad \mathbf{a}^{(k)} \in \mathbb{R}^{N_r \times d_a}. \tag{27}$$

where $N_r$ is the number of residues, and $d_a$ is the conditioning feature dimension. $\mathbf{a}^{(k)}$ serves as the static structural conditioning for the subsequent spatial-spectral autoregressive diffusion modeling module (Sec. 3.7).

SARE combines two types of input: sequence embeddings from a pretrained protein language model (ESM-2), and 3D coordinates from the structural anchor. It uses a three-step process to extract features that do not change under rotation or translation (SE(3) invariance) and to connect information across long distances in the structure.

**Stage 1: Multimodal Feature Initialization & Triangular Refinement.**   We first make residue-wise features $\mathbf{s}_i^{(0)}$ by joining ESM-2 embeddings and a learned amino-acid type embedding. For each pair, we set up pairwise features $\mathbf{z}_{ij}^{(0)}$ to show how residues are placed compared to each other. We do this by adding relative position encodings and a simple projection of the Euclidean distance $\|\mathbf{r}_i - \mathbf{r}_j\|_2$. Next, we use several *triangular self-attention blocks* (Jumper et al., 2021). These blocks use symmetric updates to pass information and keep pairwise relationships consistent. After this, we get new features $\mathbf{s}_i$ and $\mathbf{z}_{ij}$.

**Stage 2: Invariant Point Attention (IPA) for Local 3D Context.**   To incorporate fine-grained local geometry while maintaining SE(3)-invariance, we employ Invariant Point Attention (Jumper et al., 2021). Let $\mathbf{T}_i = (\mathbf{R}_i, \mathbf{t}_i)$ denote the local rigid frame for residue $i$. The IPA module takes $\mathbf{s}_i$, $\mathbf{z}_{ij}$, and $\{\mathbf{T}_i\}$ as input. It computes attention weights that depend on both feature similarity and the alignment of learned 3D point sets within these local frames:

$$\alpha_{ij} \propto \exp\Big(\frac{\mathbf{q}_i^\top \mathbf{k}_j}{\sqrt{d}} + b_{ij} - \frac{\gamma}{2} \sum_{p=1}^{P} \|\mathbf{T}_i^{-1} \circ (\mathbf{R}_j \mathbf{q}_{j,p}^p + \mathbf{t}_j) - \mathbf{q}_{i,p}^p\|^2\Big). \tag{28}$$

where $\mathbf{q}_{i,p}^p, \mathbf{q}_{j,p}^p \in \mathbb{R}^3$ are learned spatial queries and keys for point $p$ in head $h$. This mechanism updates the residue features to $\mathbf{s}_i^{\text{IPA}}$, enriching them with spatially aware contextual information.

**Stage 3: Spatial Transformer with IRFC for Global Coupling.**   In the last step, a *Spatial Transformer* layer lets all residues share information with each other. This layer uses the Diffusion Transformer from AlphaFold3 (Abramson et al., 2024). We add our Inter-Residue Frequency Coupling (IRFC) method to its attention (see Sec.subsection 3.6). The input is the IPA features $\mathbf{s}_i^{\text{IPA}}$ and the pair features $\mathbf{z}_{ij}$. We add the IRFC bias, $-\tau_h\|\mathbf{r}_i - \mathbf{r}_j\|^2$, to the attention scores. This bias makes the model focus more on close residues but still lets it connect far ones if needed. The output from this layer is the final structural conditioning vector $\mathbf{a}^{(k)}$.

## B. Algorithm of Training and Inference

This section summarizes the training and rollout procedures of BioDynaSpec in pseudocode form. Algorithm 1 describes the training pipeline: a temporal window is sampled from the trajectory, converted into normalized spectral tokens, partially revealed through progressive masking, and then processed by the structure-aware encoder, spatio-spectral planner, and diffusion denoiser. Algorithm 2 describes autoregressive rollout at test time: starting from an initial anchor conformation, the model generates one spectral window at a time in a low-to-high frequency order, reconstructs the corresponding time-domain segment via inverse rFFT, and recursively updates the anchor for the next window. For clarity, the pseudocode abstracts away some implementation-level optimizations and focuses on the main data flow and conditioning structure.

## C. Window-Size Sensitivity Study

The window size $W$ controls a fundamental trade-off in BioDynaSpec. With time step $\Delta t$, the frequency-bin spacing is $\Delta f = 1/(W\Delta t)$, so increasing $W$ improves low-frequency resolution and exposes slower collective modes more clearly.

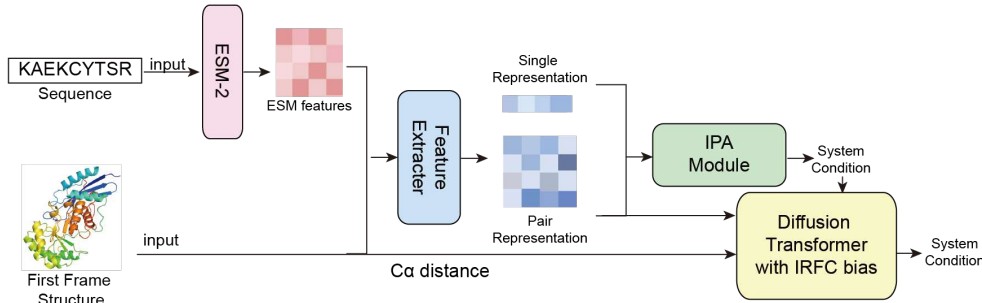

*Figure 3.* **Overview of the Structure-Aware Representation Encoder (SARE).** Given the amino-acid sequence and the first-frame anchor structure, SARE first extracts sequence features from ESM-2 and fuses them with structure-derived pairwise geometry. The resulting single-residue and pairwise representations are then processed by an IPA module to capture local SE(3)-aware geometric context, followed by a diffusion transformer with IRFC bias to further incorporate distance-aware multi-scale coupling. The final output is a structure-aware system condition used to guide subsequent spatio-spectral generation.

*Table 6.* Effect of window size $W$ on generation quality. Higher is better for validity metrics; lower is better for distributional mismatch metrics.

| Metric | $\mathbf{W = 2500}$ | $\mathbf{W = 5000}$ |
|---|---|---|
| Steric validity $\uparrow$ | **0.9514** | 0.9421 |
| Bond validity $\uparrow$ | **0.6514** | 0.6185 |
| $C_\alpha$ bond validity $\uparrow$ | **0.3517** | 0.3112 |
| PWD $\mathcal{J}$ $\downarrow$ | 0.0356 | **0.0308** |
| TICA $\mathcal{J}$ $\downarrow$ | 0.7668 | **0.6337** |
| RG $\mathcal{J}$ $\downarrow$ | 0.8092 | **0.7174** |
| RMWD $W_2$ $\downarrow$ | 3.1141 | **2.7405** |
| RMSF div. $\downarrow$ | 0.9183 | **0.4606** |

However, larger windows also weaken the assumption that a single locally coherent spectrum is sufficient within one window, especially when conformational drift accumulates over longer time spans. Conversely, smaller windows better satisfy the local-stationarity assumption, but yield coarser low-frequency coverage and introduce more stitched interfaces across rollout.

To quantify this trade-off, we compare two settings, $W = 2500$ and $W = 5000$, while keeping the rest of the pipeline fixed. In both cases, training and generation operate at the native $\Delta t = 10$ ps resolution; the 400 ps spacing used in trajectory evaluation is only post-hoc subsampling for benchmark compatibility. We report both local validity metrics (steric validity, bond validity, and $C_\alpha$ bond validity) and global/distributional metrics (PWD $\mathcal{J}$, TICA $\mathcal{J}$, RG $\mathcal{J}$, RMWD $W_2$, and RMSF divergence).

As shown in Table 6, the effect of $W$ is clearly non-monotonic. A larger window ($W = 5000$) improves global/distributional descriptors, including PWD, TICA, RG, RMWD, and RMSF divergence, consistent with finer low-frequency resolution and a better representation of slow collective modes. At the same time, it degrades local validity metrics, including steric validity, bond validity, and $C_\alpha$ bond validity, suggesting that the local-stationarity assumption becomes weaker over longer windows.

These results support our view that the independence assumption in BioDynaSpec should be interpreted as a *local approximation* rather than a globally exact factorization. Although rollout remains coupled through the recursively updated structural anchor, enlarging $W$ pushes each window further away from a locally coherent fluctuation regime. We therefore use $W = 2500$ as a practical compromise: it preserves stronger local structural fidelity while still retaining meaningful low-frequency content. Importantly, we do not claim this choice is theoretically optimal; rather, the sensitivity study indicates that window size mediates a genuine trade-off between local validity and low-frequency expressiveness.

## D. Ablation Study

We ablate the proposed Inter-Residue Frequency Coupling (IRFC) bias by removing the Gaussian distance-dependent penalty from the attention logits, while keeping all other components unchanged. We use the same evaluation protocol as §4.2.1: 82 ATLAS test proteins, 250-frame trajectories (400 ps per frame), and identical metric computation, alignment,

---

**Algorithm 1** Training

---

**Require:** Window length $W$, half-spectrum size $F = \lfloor W/2 \rfloor + 1$, diffusion steps $T_{\text{diff}} = 1000$
**Require:** Trajectory $\mathbf{X}$ ($C_\alpha$ coordinates), sequence context $\mathbf{C}_{\text{seq}}$
**Ensure:** Trained parameters of SARE $\mathcal{E}$, planner $\mathcal{P}_\theta$, and denoiser $F_\theta$

1: **repeat**
2:     Sample window index $k$
3:     $\mathbf{X}^{(k)} \leftarrow \mathbf{X}[(k-1)W + 1 : kW]$
4:     $\mathbf{x}_{\text{start}} \leftarrow \mathbf{X}^{(k)}[1]$
5:     $\mathbf{S} \leftarrow \text{rFFT}_{\text{time}}(\mathbf{X}^{(k)})$
6:     $\mathbf{X}_{\text{tok}} \leftarrow \text{ScalePerFrequency}(\text{PackReIm}(\mathbf{S}))$
7:     Sample cutoff $s \sim \mathcal{U}\{0, 1, \ldots, F\}$
8:     $\mathbf{X}_{1:s} \leftarrow \mathbf{X}_{\text{tok}}[:, 1\!:\!s, :]$
9:     $\tilde{\mathbf{X}}^{(s)} \leftarrow \text{Interp}_{\text{freq}}(\mathbf{X}_{1:s}, F)$                $\triangleright$ 1D linear interpolation along the frequency axis
10:     Sample masked frequency set $M$
11:     $\mathbf{X}_{\text{in}} \leftarrow \tilde{\mathbf{X}}^{(s)}$
12:     $\mathbf{X}_{\text{in}}[:, M, :] \leftarrow [\text{MASK}]$
13:     $\mathbf{a} \leftarrow \mathcal{E}(\mathbf{x}_{\text{start}}, \mathbf{C}_{\text{seq}})$
14:     $\mathbf{z} \leftarrow \mathcal{P}_\theta(\mathbf{X}_{\text{in}}, \mathbf{a})$
15:     Sample timestep $t \sim \mathcal{U}\{0, \ldots, T_{\text{diff}} - 1\}$
16:     $\boldsymbol{\epsilon} \sim \mathcal{N}(\mathbf{0}, \mathbf{I})$
17:     $\mathbf{x}_t \leftarrow q(\mathbf{x}_t \mid \mathbf{x}_0 = \mathbf{X}_{\text{tok}}, t, \boldsymbol{\epsilon})$
18:     $\hat{\mathbf{x}}_0 \leftarrow F_\theta(\mathbf{x}_t; t, \mathbf{z})$
19:     $\mathcal{L} \leftarrow \text{MSE}(\hat{\mathbf{x}}_0, \mathbf{X}_{\text{tok}})$
20:     Update parameters using $\nabla\mathcal{L}$

21:     *Self-forcing stage:* follow the same low-to-high autoregressive order as inference
22: **until** convergence

---

*Table 7.* Ablation on IRFC bias under the same protocol as §4.2.1 (82 ATLAS test proteins; 250-frame trajectories). We report mean±std over proteins. Lower is better for RMSE/MAE, and higher is better for Corr.

| Method | RMSE ($\downarrow$, Å) | MAE ($\downarrow$, Å) | Corr. ($\uparrow$) |
|---|---|---|---|
| BioDynaSpec | **1.540±0.784** | **0.587±0.298** | **0.996±0.004** |
| w/o IRFC bias | 33.923±6.442 | 13.214±3.293 | 0.244±0.077 |

and aggregation. For the w/o-IRFC variant, we report the best checkpoint under the same training budget. Table 7 reports mean±std over proteins.

Removing IRFC substantially degrades motion fidelity: RMSE increases from $1.540 \pm 0.784$ Å to $33.923 \pm 6.442$ Å, MAE from $0.587 \pm 0.298$ Å to $13.214 \pm 3.293$ Å, and Corr. drops from $0.996 \pm 0.004$ to $0.244 \pm 0.077$.

## E. Complexity Analysis

### E.1. Theoretical Complexity

We analyze the inference complexity of BioDynaSpec at the residue level. Let $N_r$ denote the number of residues, $W$ the window length in the time domain, and $F = \left\lfloor \dfrac{W}{2} \right\rfloor + 1$ is the number of non-negative frequency bins after the real FFT. Let $M = T/W$ be the number of windows for a trajectory of total length $T$.

For a single window, the preprocessing and reconstruction steps consist of a real FFT and inverse real FFT along the time axis. Ignoring constant factors from the three Cartesian coordinates, their complexity is $\mathcal{O}(N_r W \log W)$.

Within the spatio-spectral planner, each hybrid block alternates between: (i) *inter-residue spatial attention* across residues, whose cost scales as $\mathcal{O}(F N_r^2)$, and (ii) *intra-residue spectral attention* along the frequency axis, whose cost scales as $\mathcal{O}(N_r F^2)$. Thus, for a planner with $L$ hybrid blocks, the per-window planner complexity is

$$\mathcal{O}\big(L\big(F N_r^2 + N_r F^2\big)\big). \tag{29}$$

---

**Algorithm 2** Inference / Rollout

---

**Require:** Number of windows $M$, sampling steps $K = 50$, initial anchor $\mathbf{x}_{\text{start}}^{(1)}$, sequence context $\mathbf{C}_{\text{seq}}$
**Ensure:** Generated trajectory $\hat{\mathbf{X}}$

1: $\mathbf{x}_{\text{start}} \leftarrow \mathbf{x}_{\text{start}}^{(1)}$
2: $\hat{\mathbf{X}} \leftarrow [\,]$
3: **for** $k = 1$ to $M$ **do**
4:      $\mathbf{a} \leftarrow \mathcal{E}(\mathbf{x}_{\text{start}}, \mathbf{C}_{\text{seq}})$
5:      $\mathbf{tokens} \leftarrow \mathbf{0} \in \mathbb{R}^{N_r \times F \times D}$
6:      **for** $f = 0$ to $F - 1$ **do**
7:          $\mathbf{z} \leftarrow \mathcal{P}_\theta(\mathbf{tokens}, \mathbf{a})$
8:          $\mathbf{index} \leftarrow (f / \max(F - 1, 1)) \cdot \mathbf{1}$
9:          $\mathbf{tokens}_{\text{new}} \leftarrow \texttt{DiffusionSample}(F_\theta, \mathbf{z}, \mathbf{index}, K)$
10:          **if** $f < F - 1$ and $f > 0$ **then**
11:             $\mathbf{tokens}_{\text{new}} \leftarrow \text{Interp}_{\text{freq}}(\mathbf{tokens}_{\text{new}}[:, 1 : (f+1), :], F)$         ▷ 1D linear interpolation along the frequency axis
12:          $\mathbf{tokens} \leftarrow \mathbf{tokens}_{\text{new}}$
13:      $\mathbf{spectrum} \leftarrow \texttt{recover\_spectrum\_from\_tokens}(\mathbf{tokens})$
14:      $\mathbf{spectrum} \leftarrow \texttt{inverse\_scale\_per\_frequency}(\mathbf{spectrum})$
15:      $\hat{\mathbf{S}} \leftarrow \texttt{merge\_to\_complex\_half\_spectrum}(\mathbf{spectrum})$
16:      $\hat{\mathbf{X}}^{(k)} \leftarrow \text{iRFFT}_{\text{time}}(\hat{\mathbf{S}}, W)$
17:      $\mathbf{x}_{\text{start}} \leftarrow \hat{\mathbf{X}}^{(k)}[W]$
18:      Append $\hat{\mathbf{X}}^{(k)}$ to $\hat{\mathbf{X}}$
19: **return** $\hat{\mathbf{X}}$

---

Combining the transform and planner terms, the total inference complexity per window is

$$\mathcal{O}\big(N_r W \log W + L(F N_r^2 + N_r F^2)\big), \tag{30}$$

up to the cost of the diffusion denoising steps. Since rollout proceeds window by window, the total cost over a trajectory is linear in the number of windows:

$$\mathcal{O}\Big(M\big[N_r W \log W + L(F N_r^2 + N_r F^2)\big]\Big). \tag{31}$$

This decomposition shows that FFT/iRFFT is not the dominant asymptotic term in our system. The main cost arises from the spatio-spectral planner and diffusion sampling, while the Fourier transform contributes a comparatively modest preprocessing/reconstruction overhead.

For reference, the baseline complexity can be summarized at a high level as: MDGEN: $\mathcal{O}(KTN^2)$; ProAR: $\mathcal{O}(HNk)$, where $K$ denotes diffusion sampling steps, $T$ the rollout horizon, $N$ the system size, $H$ the prediction horizon, and $k$ the number of frame parameters or mixture components used in ProAR.

### E.2. Empirical Wall-Clock Time and Parameter Count

To complement the asymptotic analysis, we report empirical wall-clock time under the same trajectory-generation setting used in the main paper: ATLAS, 250 frames per protein, evaluated on a single NVIDIA H200 GPU. We also report the parameter count of BioDynaSpec.

Two observations are immediate from Table 8. First, BioDynaSpec operates in the same overall runtime regime as MDGEN, while being substantially faster than ProAR at the full-pipeline level. Second, for BioDynaSpec, the iRFFT reconstruction cost is negligible compared with the network inference cost ($0.054 \pm 0.028$ s vs. $15.867 \pm 12.849$ s), which empirically confirms the theoretical analysis above: the Fourier-domain transform itself does not dominate runtime.

In other words, the practical trade-off in BioDynaSpec is not that FFT/iRFFT overwhelms the computational budget. Rather, the dominant cost remains the autoregressive planner and diffusion denoising, while the frequency-domain formulation provides a more structured low-to-high generation process with only a small transform overhead. Overall, BioDynaSpec is approximately $57\times$ faster than ProAR under the measured full-pipeline setting, while remaining close to MDGEN in wall-clock runtime.

*Table 8.* Empirical wall-clock time for trajectory generation (ATLAS, 250 frames/protein, single H200 GPU). For BioDynaSpec and ProAR, we additionally report the available decomposition into major runtime components.

| Model | Time | Decomposition |
|---|---|---|
| MDGEN | 11.05 s/protein | — |
| ProAR | $917.302 \pm 606.998$ s/protein | $24.781 \pm 6.359$ s (network) |
| | | $892.521 \pm 600.973$ s (structure relaxation) |
| BioDynaSpec | $15.921 \pm 12.859$ s/protein | $15.867 \pm 12.849$ s (network) |
| | | $0.054 \pm 0.028$ s (iRFFT) |

## F. Software and Data

The full BioDynaSpec source code, including the model architecture, training pipeline, and inference scripts, is available at https://github.com/Linmj-Judy/BioDynaSpec.git.

The raw ATLAS protein dynamics dataset used in this work is publicly available at https://www.dsimb.inserm.fr/ATLAS. The frame-extraction scripts used to preprocess the raw molecular dynamics trajectories are available at https://github.com/Linmj-Judy/BioDynaSpec/tree/main/runner/data_prepare/0_extract_frames, and the training-data preparation scripts are available at https://github.com/Linmj-Judy/BioDynaSpec/tree/main/runner/data_prepare/2_prepare_data.

