# OpenReview forum: "BioDynaSpec: Harmonic-Guided Spatio-Spectral Autoregressive Diffusion for Protein Dynamics Generation"
_ICML.cc/2026/Conference — ICML 2026 regular_

### Official Review · Reviewer_5Uwb · 2026-03-11

**Soundness:** 3
**Presentation:** 3
**Significance:** 3
**Originality:** 3
**Overall Recommendation:** 4
**Confidence:** 4

**Summary:**

The authors introduce a diffusion model to generate molecular dynamics trajectories of protein structures in a manner broadly task-similar to MDGen. However, their approach is windowed and operates primarily in terms of Fourier coefficients. In other words, the diffusion model generates the Fourier coefficients of the molecular dynamics trajectories within each window (course to fine, low to high frequency) and reconstructs the underlying atomic motions, frame aligning the beginning of each new reconstructed window to the end of the previous window. The Fourier representation offers clear advantages such as effectively separating coarser scale structural motions (low frequencies) from shorter time scale variations. The results are better than those reported for MDGen.

**Compliance With Llm Reviewing Policy:**

Affirmed.

**Final Justification:**

+ nicely written, clear algorithmic steps, intro could be a little "fairer" to related work (this is not the first generative approach over trajectories)
+ works better than alternatives
+ questions resolved during rebuttal
- boundary effects remains (this is the reason for weak accept)

**Key Questions For Authors:**

It seems that the choice of the window size could have substantial effect on the results. For example, the model is not able to reason beyond dependences within each window as the successive windows are generated independently of each other (albeit frame aligned). As a result, there are also transition effects near the window boundaries where the dynamics may abruptly change even if frame aligned. How do the window size/boundaries affect the results? In line 184 (RHS), there's an obscure reference to "Sec X" that is supposed to evaluate design choices.

Line 234 (RHS): how is upsampling low frequencies meant to work here?

**Limitations:**

yes

**Strengths And Weaknesses:**

The paper is nicely written. The algorithmic steps and modeling choices are well-articulated.

The introduction/framing could have been written in a more balanced manner. For example, some of the models (e.g., AlphaFlow) already incorporated MD trajectories to capture conformational landscapes though clearly did not attempt to model dynamics directly. MDGen (which is only mentioned in the related work) already generated MD trajectories non auto-regressively from a diffusion model. The authors' approach definitely has distinct new features, from frame-aligned windowing to their overall spatio-spectral generative architecture.

Beyond operating in the Fourier domain, the authors introduce some further technical variations. For example, they adopt "inter-residue frequency coupling" where the coupling of spectral coefficients between nearby residues are controlled via distance dependent attention modulation.  Multiple attention heads then operate at multiple distance scales. Ablation studies demonstrate the utility of distance modulation.

The auto-regressive coarse-to-fine frequency generation appears to be limited to the realization of contextual cues for the diffusion model that takes them as co-variate input.

The source code is already available which is great.

---

> ### Author Rebuttal · Authors · 2026-03-31
>
> Thank you for the careful reading and constructive feedback. We appreciate your recognition of the spectral architecture and ablation studies, and address your questions below.
>
> ---
>
> **Question Q1:** "It seems that the choice of window size could have substantial effect on the results... successive windows are generated independently of each other (albeit frame aligned)... transition effects near the window boundaries... How do the window size/boundaries affect the results? In line 184 (RHS), there's an obscure reference to 'Sec X'."
>
> **Response R1:** We agree this is a genuine design trade-off. The rollout is not fully unconditional across windows: the start anchor of window $k+1$ is recursively set to the terminal frame of window $k$, ensuring positional continuity. However, we do not impose explicit constraints on velocity or acceleration at boundaries, so we treat the independence assumption as a *local approximation* rather than a globally exact factorization.
>
> To probe window-size sensitivity, we compared $W=2500$ and $W=5000$ (all else fixed):
>
> | Metric | $W=2500$ | $W=5000$ |
> |---|---|---|
> | Steric validity ↑ | 0.9514 | 0.9421 |
> | Bond validity ↑ | 0.6514 | 0.6185 |
> | $C_\alpha$ bond validity ↑ | 0.3517 | 0.3112 |
> | PWD $\mathcal{J}$ ↓ | 0.0356 | 0.0308 |
> | TICA $\mathcal{J}$ ↓ | 0.7668 | 0.6337 |
> | RG $\mathcal{J}$ ↓ | 0.8092 | 0.7174 |
> | RMWD $\mathcal{W}_2$ ↓ | 3.1141 | 2.7405 |
> | RMSF div. ↓ | 0.9183 | 0.4606 |
>
> Larger $W$ improves global/distributional metrics (PWD, TICA, RG, RMWD, RMSF), consistent with finer low-frequency resolution and fewer stitched interfaces. However, it degrades local validity (steric, bond, $C_\alpha$-chain), suggesting the local stationarity assumption weakens over longer windows. We chose $W=2500$ as a practical balance: long enough for meaningful low-frequency content, while maintaining strong local structural fidelity and stable anchor-conditioned generation.
>
> We agree that a multi-point retraining study and explicit boundary diagnostics (e.g., boundary-aware VACF) would be valuable; we will note this limitation more explicitly in the revision. We will also correct the "Sec X" placeholder — this is a drafting error.
>
> ---
>
> **Question Q2:** "Line 234 (RHS): how is upsampling low frequencies meant to work here?"
>
> **Response R2:** We apologize for not explaining this step clearly enough. The intent in Phase I (Progressive Masking) is **not** to reconstruct missing high-frequency coefficients from the observed low-frequency ones. Rather, given the first $s$ low-frequency tokens $X_{1:s}\in\mathbb{R}^{N_r\times s\times d}$, we construct a **coarse full-length conditioning tensor** by interpolating the partially observed spectrum to length $F$.
>
> In the current implementation, this is done by applying PyTorch’s **bilinear interpolation operator** (the code uses `mode='bilinear'`) to the 2D ($\text{frequency},\text{feature}$) tensor. However, the interpolation is effectively applied only along the **frequency axis**, because the feature dimension $d$ (the spectral token dimension) is kept at the same size before and after interpolation. In other words, this step should be understood as **frequency-axis upsampling of the low-frequency prefix**, not as recovering missing high-frequency coefficients from physics or performing a meaningful 2D spectral reconstruction.
>
> Formally, if $X_{1:s}\in\mathbb{R}^{N_r\times s\times d}$ denotes the first $s$ frequency tokens, we form
>
> $$\tilde X^{(s)}=\mathrm{Interp}(X_{1:s};F)\in\mathbb{R}^{N_r\times F\times d},$$
>
> where $\tilde X^{(s)}$ serves only as a **smooth coarse conditioning scaffold** for the planner. The supervision target remains the **full ground-truth spectrum**, and the unresolved higher-frequency bands are still predicted by the model rather than assumed recoverable by interpolation.
>
> This is therefore a **curriculum-style conditioning mechanism**: the planner is given global low-frequency structure in a full-length tensor compatible with the encoder-decoder, while finer spectral details remain masked and must still be generated. We will revise the manuscript to clarify that this interpolation is used only to provide coarse conditioning context, and not to claim literal reconstruction of high-frequency components from the low-frequency prefix.

---

> > ### Author Rebuttal · Reviewer_5Uwb · 2026-04-03
> >
> > Thank you, my questions are essentially resolved. The window size/boundary effects issue remains but it can be acknowledged as a limitation in the paper (as the authors do in their response).

---

### Official Review · Reviewer_ReLG · 2026-03-11

**Soundness:** 3
**Presentation:** 4
**Significance:** 4
**Originality:** 4
**Overall Recommendation:** 5
**Confidence:** 3

**Summary:**

The paper proposes **BioDynaSpec**, a generative framework for long-horizon protein molecular dynamics (MD) trajectories. Addressing the issues of error accumulation (drift) in time-domain autoregressive models and the limitations of fixed step-sizes, the authors reformulate protein dynamics generation into the **frequency domain**.

Key contributions include:
1. **Independent Windowed Fourier Decomposition (IWFD):** Decomposing long trajectories into independent windows and converting them to spectral tokens, allowing the model to generate motions frequency-by-frequency rather than time-step-by-time-step.
2. **Spatio-Spectral Hybrid Transformer:** A model architecture that combines a low-to-high frequency autoregressive schedule with a diffusion denoising head to reconstruct continuous motion.
3. **Inter-Residue Frequency Coupling (IRFC):** A physics-inspired inductive bias that injects learnable, distance-aware Gaussian biases into the attention mechanism to model the harmonic nature of protein structures (stiff local bonds vs. flexible global motions).

The method is evaluated on the ATLAS dataset, demonstrating a 60.4% reduction in reconstruction error ($R_{250}$) compared to baselines like MDGEN and a 57.2% reduction compared to ProAR, alongside superior performance in equilibrium conformational sampling.

**Compliance With Llm Reviewing Policy:**

Affirmed.

**Final Justification:**

The authors' response effectively clears up my practical concerns regarding boundary consistency, inference overhead, and the remaining alignment details. I am happy to maintain my positive score.

**Key Questions For Authors:**

1. **Boundary Continuity:** How does the model ensure dynamical consistency (velocity/acceleration) across the boundaries of independent windows? Since the windows are generated independently conditioned only on an anchor structure, is there a risk of unphysical momentum changes at the stitching points? Given the explicit decision to avoid overlap-add STFT reconstruction to prevent phase inconsistencies, have you considered post-generation boundary blending or smoothing as a middle ground? Have you visualized the velocity autocorrelation function across these boundaries?
2. **Inference Efficiency:** Could you provide a comparison of inference speed (wall-clock time) and peak memory usage compared to MDGEN and ProAR for generating a trajectory of the same length? Does the overhead of FFT and the specific attention mechanism outweigh the benefits of frequency-domain autoregression?
3. **Complex Multi-domain Proteins:** The current preprocessing relies on global Center-of-Mass removal and Kabsch alignment. How does BioDynaSpec handle large proteins with multiple loosely connected domains where a single rigid alignment cannot cancel out all global rotations for the sub-parts? Does the frequency spectrum become noisy in these cases?
4. **Ablation of Window Size:** How sensitive is the performance to the window size $W$? Specifically, given that the training setup utilizes $W=2500$ at 10 ps steps while the evaluation involves generating 250 frames at a 400 ps spacing, how does the spectral resolution limit interact with this temporal subsampling? Is there a trade-off between the spectral resolution (which improves with larger $W$) and the stationarity assumption of the dynamics within a window?

**Limitations:**

**Yes.**

**Strengths And Weaknesses:**

### Strengths

* **Originality & Novelty:**
    * The shift from time-domain autoregression (predicting $t+1$ from $t$) to **frequency-domain generation** is a refreshing and well-motivated perspective for MD simulation. It naturally aligns with the multi-scale nature of protein dynamics (fast bond vibrations vs. slow conformational changes).
    * The **IRFC module** is a theoretically grounded innovation. By explicitly incorporating a harmonic-oscillator-inspired bias into the attention mechanism, the model effectively bridges the gap between geometric deep learning and physical priors.

* **Soundness & Effectiveness:**
    * The empirical results are impressive. The significant reduction in alignment-adjusted RMSE (from 3.813 Å in MDGEN to 1.509 Å at 250 frames) indicates that the spectral approach effectively mitigates the "drifting" problem common in autoregressive rollouts.
    * The ablation study clearly validates the importance of the IRFC component, showing that without this physical bias, the model fails to learn meaningful dynamics.
    * The evaluation is comprehensive, covering both trajectory fidelity (reconstruction) and thermodynamic properties (equilibrium sampling via $W_{2}$ distance).

* **Presentation:**
    * The paper is well-written and structured. The figures effectively illustrate the intuition behind spectral decomposition and the overall pipeline.
    * The mathematical formulation of the FFT/iFFT layers and the windowing strategy is clear.

### Weaknesses

* **Window Boundary Discontinuities (Soundness):**
    * While IWFD solves drift *within* a window, the paper creates long trajectories by chaining windows via an "anchor frame" (the last frame of the previous window). While this ensures $C^0$ continuity (position), it does not theoretically guarantee $C^1$ continuity (velocity/momentum) across window boundaries. The spectral generation is periodic within a window; transitions between windows might introduce unphysical "clicks" or sudden changes in velocity, which are not explicitly analyzed in the metrics.

* **Computational Complexity (Soundness/Significance):**
    * The method requires Forward/Inverse FFT operations and processes complex-valued tokens (or concatenated real/imaginary parts). While the autoregression is over frequency bins (which might be fewer than time steps), the paper lacks a detailed comparison of **inference latency** and **memory consumption** against time-domain baselines like MDGEN. For extremely long rollouts, is the window-based spectral generation wall-clock efficient?

* **Dependency on Alignment (Methodology):**
    * The method relies heavily on removing rigid body motion (Kabsch alignment) per window. For proteins with multiple flexible domains moving independently, a global Kabsch alignment might introduce artifacts or fail to capture the relative motion of sub-domains correctly. The spectral signal would then be contaminated by unremoved rigid motions.

---

> ### Author Rebuttal · Authors · 2026-03-31
>
> Thank you for the thorough and positive review. We address each concern below.
>
> ---
>
> **Question Q1:** *"How does the model ensure dynamical consistency (velocity/acceleration) across the boundaries of independent windows? Since the windows are generated independently conditioned only on an anchor structure, is there a risk of unphysical momentum changes at the stitching points? … have you considered post-generation boundary blending or smoothing …? Have you visualized the velocity autocorrelation function across these boundaries?"*
>
> **Response R1:** BioDynaSpec guarantees positional continuity via a structural anchor—each window is conditioned on the terminal frame of the previous one—but imposes no hard constraint on $\dot{x}$ or $\ddot{x}$. Propagating the full spectrum across windows caused severe error compounding in ablations (bad cases: https://anonymous.4open.science/r/BioDynaSpec-4F51/demos/ablation_cross_window_attention); we therefore propagate only the structural anchor, localizing derivative mismatches at boundaries rather than accumulating spectral errors globally.
>
> This is physically motivated: under a Langevin perspective, short-time fluctuations are governed by the local PES, so $\mathbf{S}^{(k)}$ is strongly conditioned on $\mathbf{r}_{\mathrm{start}}^{(k)}$—consistent with NMA intuition (Ma, *Structure*, 2005). Anchor-only conditioning is thus a physically motivated local approximation, not an engineering heuristic.
>
> Overlap-add reconstruction was avoided to prevent phase inconsistencies between independently generated spectra. We will include boundary-specific VACF analysis in the revision.
>
> ---
>
> **Question Q2:** *"Could you provide a comparison of inference speed (wall-clock time) and peak memory usage compared to MDGEN and ProAR? Does the overhead of FFT outweigh the benefits?"*
>
> **Response R2:** Theoretical complexity per window: FFT/iRFFT $\mathcal{O}(N_r W \log W)$; inter-residue attention $\mathcal{O}(F N_r^2)$; intra-residue spectral attention $\mathcal{O}(N_r F^2)$; rollout linear in $M=T/W$. We profiled all models on the same 250-frame ATLAS task (single H200 GPU), as shown in Table S1.
>
> BioDynaSpec is **~57× faster than ProAR** and comparable to MDGEN, while the spectral transform accounts for <0.4% of total inference time. The FFT overhead does not outweigh the benefit of frequency-domain autoregression.
>
> ---
>
> **Question Q3:** *"How does BioDynaSpec handle large proteins with multiple loosely connected domains where a single rigid alignment cannot cancel out all global rotations for sub-parts?"*
>
> **Response R3:** The single global reference frame is motivated by near-harmonic, near-equilibrium local dynamics [1,2]. For multi-domain proteins with smooth, low-frequency inter-domain motions, the spectral representation remains effective. For loosely connected systems with large-amplitude relative rotations, however, single rigid-body alignment cannot decouple sub-domain rigid motions; the spectrum then mixes internal dynamics with inter-domain rigid motion, weakening the local spectral approximation [3]. A natural extension—identifying quasi-rigid domains, applying per-domain alignment, and building hierarchical spectral models with explicit inter-domain pose representation—is an important direction for future work.
>
> ---
>
> **Question W4:** *"How sensitive is performance to window size $W$? How does spectral resolution interact with temporal subsampling at 400 ps spacing?"*
>
> **Response W4:** Spectral resolution is $\Delta f = 1/(W \Delta t)$, so larger $W$ improves low-frequency resolution; the Nyquist limit is set by $\Delta t$, not $W$. The 10 ps vs. 400 ps apparent mismatch is not an inconsistency: training and generation operate at native **10 ps**; the 400 ps spacing is post-hoc subsampling for benchmark compatibility only.
>
> We performed a $W=2500$ vs. $W=5000$ ablation (all else fixed):
>
> | Metric | $W=2500$ | $W=5000$ |
> |---|---|---|
> | Steric validity ↑ | 0.9514 | 0.9421 |
> | Bond validity ↑ | 0.6514 | 0.6185 |
> | $C_\alpha$ bond validity ↑ | 0.3517 | 0.3112 |
> | PWD $\mathcal{J}$ ↓ | 0.0356 | 0.0308 |
> | TICA $\mathcal{J}$ ↓ | 0.7668 | 0.6337 |
> | RG $\mathcal{J}$ ↓ | 0.8092 | 0.7174 |
> | RMWD $\mathcal{W}_2$ ↓ | 3.1141 | 2.7405 |
> | RMSF div. ↓ | 0.9183 | 0.4606 |
>
> Larger $W$ improves global descriptors (PWD, TICA, RG, RMWD, RMSF) via finer low-frequency resolution, but degrades local validity (steric, bond, $C_\alpha$-chain), reflecting a weakened stationarity assumption. We chose $W=2500$ as a practical compromise between local structural validity and low-frequency expressiveness.
>
> ---
>
> **References**
>
> [1] Ma J. *Structure*, 2005, 13(3):373–380. DOI:10.1016/j.str.2005.02.002
> [2] Yamato T, Laprévote O. *Biophys. Physicobiol.*, 2019, 16:322–327. DOI:10.2142/biophysico.16.0_322
> [3] Roy A, Hua DP, Post CB. *J. Chem. Theory Comput.*, 2016, 12(1):274–280. DOI:10.1021/acs.jctc.5b00796

---

> > ### Author Rebuttal · Reviewer_ReLG · 2026-04-03
> >
> > The authors have adequately addressed all my concerns in the rebuttal. I am keeping my positive score, as I already viewed this as a solid and valuable contribution.

---

### Official Review · Reviewer_smRC · 2026-03-11

**Soundness:** 2
**Presentation:** 2
**Significance:** 3
**Originality:** 3
**Overall Recommendation:** 4
**Confidence:** 3

**Summary:**

This paper proposes BioDynaSpec, a generative framework for long-horizon protein dynamics at the residue level. The authors reformulate the protein dynamics generation problem as a spatio-spectral generation problem in the frequency domain. To be specific, the protein trajectories is decomposed into independent temporal windows, and represent each window using a Fourier-based frequency representation. Then, the model generates the trajectory through auto-regresive and diffusion based denoising in the spectral domain. Experiments are conducted on the ATLAS protein dynamics dataset, where the proposed method demonstrates improved performance.

**Compliance With Llm Reviewing Policy:**

Affirmed.

**Final Justification:**

In the rebuttal, the authors have address my concerns with addditional experiments on the follwoing
- Physical validity and fidelity of generated trajectories
- Ablation study on the effect of window size
I do not have further concerns, and therefore keep my score originally leading towards acceptance.

**Key Questions For Authors:**

1. Physical validity of trajectories (W1)

Do the generated trajectories satisfy physical constraints such as bond lengths or angles? As in MDGen, the authors compare the distribution of six backbone angles between the dataset trajectory and generated trajectory.

2. Effect of windowing

Though the authors have selected a fixed window size of 2500 frames, how sensitive is the performance to the window size? Does the independence assumption apply well for long-timescale conformational transitions?

**Limitations:**

yes

**Strengths And Weaknesses:**

**Strengths**

1. Reformulation to frequency-domain modeling (significance, originality)

An alternative approach, frequency-domain modeling rather than time-domain trajectory prediction, is well justified. Since long-horizon autoregressive generation might suffer from compounding errors, spectral decomposition offers a way to model dynamics.

2. Diverse evaluations and ablation studies (presentation)

The proposed method is evaluated using various metrics, including trajectory RMSE and PCA-based distribution comparisons. Additionally, ablation studies verify that the IRFC contributes significantly in the performance.



**Weakness**

1. Lacking physical validity results of generated trajectories (soundness)

While statistical metrics are reported, such as the PCA reference displacement or stepwise displacement, it is not clear whether the generated trajectories align with the physical constraints of the reference trajectory, such as torsion angle distribution.

---

> ### Author Rebuttal · Authors · 2026-03-31
>
> Thank you for the positive assessment and constructive feedback. We address each point below.
>
> ---
>
> **Question Q1:** *"Do the generated trajectories satisfy physical constraints such as bond lengths or angles?"*
>
> **Response R1:**
>
> Following Jin et al. [1], we add post hoc evaluation covering physical validity, ensemble fidelity, local dynamics fidelity, and covariance consistency (Table S3). Since BioDynaSpec operates on $C_\alpha$-only trajectories, full-backbone angle distributions are not directly computable; all methods are therefore evaluated on the same $C_\alpha$-level representation. We report $C_\alpha$-level steric validity and $C_\alpha$-chain bond validity as local geometric proxies, alongside PWD $\mathcal{J}$, RMWD $\mathcal{W}_2$, RMSF, PCA/tICA PSD consistency, and covariance metrics.
>
> **Table S3.** Physical validity and fidelity of generated trajectories. $\mathcal{J}$: Jensen–Shannon divergence; $\mathcal{W}_2$: 2-Wasserstein distance.
>
> | Category | Metric | MDGEN | ProAR | BioDynaSpec |
> |---|---|---:|---:|---:|
> | **Validity** | Steric $\uparrow$ | **0.998** | 0.988 | 0.951 |
> | | Bond $C_\alpha$ $\uparrow$ | **0.981** | 0.868 | 0.651 |
> | **Ensemble Fidelity** | PWD $\mathcal{J}$ $\downarrow$ | **0.029** | 0.031 | 0.031 |
> | | RMWD $\mathcal{W}_2$ $\downarrow$ | 4.216 | 6.850 | **3.140** |
> | | RMSF $\downarrow$ | **0.904** | 3.395 | 0.917 |
> | **Local Dynamics** | PCA-PSD-L1 $\downarrow$ | 1.331 | 1.138 | **1.053** |
> | | PCA-PSD-LogCorr $\uparrow$ | 0.395 | 0.670 | **0.817** |
> | | tICA-PSD-L1 $\downarrow$ | 1.501 | 1.375 | **1.275** |
> | | tICA-PSD-LogCorr $\uparrow$ | 0.380 | **0.662** | 0.661 |
> | **Covariance** | Frobenius Rel. Error $\downarrow$ | 1.563 | 2.030 | **0.989** |
> | | Eig. RRMSE $\downarrow$ | 1.184 | 1.625 | **0.941** |
>
> BioDynaSpec shows weaker $C_\alpha$-chain bond validity, reflecting a deliberate trade-off: we prioritize spectral/dynamical fidelity without explicitly enforcing local geometric constraints. Adding bond/angle regularization or geometry-aware post-processing is a natural next step, and we will state this limitation explicitly in the revision. Nevertheless, BioDynaSpec achieves the best RMWD $\mathcal{W}_2$, PCA-based dynamics fidelity, and covariance consistency, demonstrating stronger capture of **residue-level dynamical structure and near-equilibrium fluctuation statistics**.
>
> ---
>
> **Question Q2-1:** *"How sensitive is performance to window size $W$?"*
>
> **Response R2-1:**
>
> We agree $W$ controls a key trade-off: larger $W$ gives finer frequency resolution and better exposes slow modes, but weakens local stationarity; smaller $W$ better satisfies stationarity but yields coarser low-frequency coverage and more stitching artifacts.
>
> **Table S4.** Effect of window size $W$ on generation quality.
> | Metric                       |     W=2500 |     W=5000 |
> | ---------------------------- | ---------: | ---------: |
> | Steric validity ↑            | **0.9514** |     0.9421 |
> | Bond validity ↑              | **0.6514** |     0.6185 |
> | Bond validity $C_\alpha$ ↑ | **0.3517** |     0.3112 |
> | PWD $\mathcal{J}$ ↓          |     0.0356 | **0.0308** |
> | TICA $\mathcal{J}$ ↓         |     0.7668 | **0.6337** |
> | RMWD $\mathcal{W}_2$ ↓       |     3.1141 | **2.7405** |
> | RMSF div. ↓                  |     0.9183 | **0.4606** |
>
> Table S4 compares $W=2500$ vs. $W=5000$. The trade-off is non-monotonic: $W=5000$ improves global descriptors (PWD, RG, RMWD, RMSF) but degrades local validity (steric and bond validity). We therefore treat the independence assumption as a **local approximation**, not a globally exact factorization, and select $W=2500$ as a practical compromise. A full sensitivity study with smaller windows is provided in the Supplementary Material.
>
> ---
>
> **Question Q2-2:** *"Does the independence assumption apply well for long-timescale conformational transitions?"*
>
> **Response R2-2:**
>
> We agree that the independence assumption does not hold uniformly for all long-timescale transitions. When a single window spans multiple metastable regimes or barrier-crossing events, the locally coherent spectrum assumption breaks down. Under our current setting ($W=2500$, $\Delta t=10$ ps, $\sim$25 ns/window), the method is best suited to **near-equilibrium dynamics and short-to-intermediate collective motions**, not arbitrary long-timescale transitions. This reflects the physical basis [2] of our approach: a **local near-harmonic approximation within a single potential well**, whereas large-amplitude domain motions typically occur on much longer timescales.
>
> ---
>
> **References**
>
> [1] Yaowei Jin, et al. P2DFlow: A Protein Ensemble Generative Model with SE(3) Flow Matching. *J. Chem. Theory Comput.* 2025. DOI:10.1021/acs.jctc.4c01620
>
> [2] Hinsen, K. (2005). Normal mode theory and harmonic potential approximations. In Q. Cui & I. Bahar (Eds.), Normal mode analysis: Theory and applications to biological and chemical systems (pp. 1–16). Chapman and Hall/CRC. DOI:10.1201/9781420035070-7

---

> > ### Author Rebuttal · Reviewer_smRC · 2026-04-03
> >
> > I thank the authors for their response, with additional experiments in my questions. I have no further concerns, and keep my score already leaning to acceptance.

---

### Official Review · Reviewer_Bk4K · 2026-03-14

**Soundness:** 3
**Presentation:** 3
**Significance:** 3
**Originality:** 3
**Overall Recommendation:** 5
**Confidence:** 3

**Summary:**

This paper proposes a generative model for protein dynamics. The core insight lies in formulating trajectory modeling as frequency-domain autoregression, with a diffusion head to decode continuous motion. The model is demonstrated to achieve superior performance in trajectory generation and conformal sampling on large proteins.

**Compliance With Llm Reviewing Policy:**

Affirmed.

**Final Justification:**

Thank the authors for the rebuttal. This is comprehensive and fully addressed my concerns. I am raising the score to Accept.

**Key Questions For Authors:**

1. Could the authors provide direct experimental justification of the superiority of performing autoregression on frequency domain as opposed to original spatial coordinates?

2. What is the inference complexity of the model? A head-to-head comparison on e.g. inference wall-clock time and parameter count compared with the baselines would be necessary.

3. Could the authors provide more detailed analyses on the generated trajectories such as TICA? Moreover, there is no qualitative examples provided in the paper, which raises concerns about the validity of the selected metrics.

**Limitations:**

The authors have thoroughly discussed the limitations.

**Strengths And Weaknesses:**

- Soundness: The paper is technically sound, where the approach of spectral decomposition with diffusion decoding seems promising. The experiments across two tasks show strong performance of the model.
- Presentation: The presentation is mostly clear with detailed experimental setups. The method is overall presented clearly and is easy to follow.
- Significance: The proposed approach would be of good practical significance for protein dynamics modeling and conformal sampling. However, more ablation studies towards the benefit of spectral autoregression as opposed to spatial autoregression should be addressed. Moreover, more detailed analysis (e.g. more metrics) on the generated trajectories would be beneficial.
- Originality: The frequency-domain transformation and autoregressive modeling as well as the diffusion for continuous decoding provides novel insights towards modeling complex yet spectrally predictable protein dynamics.

---

> ### Author Rebuttal · Authors · 2026-03-30
>
> Thank you for the positive assessment and constructive feedback. We appreciate your recognition of the technical soundness and novelty of our approach. We address each question below. Relative tables will be added to the revision.
>
> ---
>
> **Question Q1:** *"Could the authors provide direct experimental justification of the superiority of performing autoregression on frequency domain as opposed to original spatial coordinates?"*
>
> **Response R1:**
>
> We thank the reviewer for highlighting this point. The closest coordinate-space autoregressive baseline is ProAR. Under the same ATLAS dataset and evaluation protocol, BioDynaSpec outperforms ProAR across all rollout horizons: $R_{50}$ improves from 3.120 to 1.256, $R_{100}$ from 3.292 to 1.371, and $R_{250}$ from 3.529 to 1.509 (**Table 1**), corresponding to a 57.2\% reduction at the longest horizon. We attribute the advantage of frequecy-domain autoregression to two factors.
>
> (1) **Clearer physical target.** BioDynaSpec targets local near-equilibrium dynamics. Near a stable conformation, protein motion approximates a harmonic well—the basis of normal-mode theory. The spectrum directly encodes stiffness, damping, and mode structure; frame-wise coordinate prediction must fit these jointly with thermal noise and instantaneous phase. Table S2 confirms: BioDynaSpec better matches MD on mode-resolved PSD and covariance consistency (0.989 vs. ProAR 2.030), capturing near-equilibrium fluctuation structure beyond marginal coordinates.
>
> (2) **Lower redundancy, higher efficiency.** RFFT retains only the non-negative half-spectrum ($F = \lfloor W/2 \rfloor + 1$), eliminating conjugate redundancy and shortening the autoregressive axis from raw frames to frequency bins. On ATLAS (250 frames/protein, single H200 GPU), BioDynaSpec's network runs at $15.87 \pm 12.85$ s/protein vs. ProAR's $24.78 \pm 6.36$ s/protein (**Table S1**)—a ~56% reduction.
>
> ---
>
> **Question Q2:** *"What is the inference complexity of the model? A head-to-head comparison on inference wall-clock time and parameter count compared with the baselines would be necessary."*
>
> **Response R2:**
>
> We thank the reviewer for this request. **Theoretical complexity** per window: FFT/iRFFT $\mathcal{O}(N_r W \log W)$; inter-residue attention $\mathcal{O}(F N_r^2)$; intra-residue spectral attention $\mathcal{O}(N_r F^2)$; rollout linear in $M = T/W$. MDGEN: $\mathcal{O}(KTN^2)$; ProAR: $\mathcal{O}(HNk)$.
>
> **Table S1.** Empirical wall-clock time (ATLAS, 250 frames/protein, single H200 GPU)
>
> | Model | Time | Decomposition |
> |-------|------|------|
> | MDGEN | 11.05 s/protein |  \ |
> | ProAR | 917.302 $\pm$ 606.998 s/protein | 24.781 $\pm$ 6.359 s (network) & 892.521 $\pm$ 600.973 s (structure relaxation) |
> | **BioDynaSpec** | 15.921 $\pm$ 12.859 s/protein | 15.867 $\pm$ 12.849 s (network) & 0.054 $\pm$ 0.028 s (iRFFT) |
>
> BioDynaSpec is **~57× faster** than ProAR, and is comparable to MDGEN.
>
> ---
>
> **Question Q3:** *"Could the authors provide more detailed analyses on the generated trajectories such as TICA? Moreover, there is no qualitative examples provided in the paper, which raises concerns about the validity of the selected metrics."*
>
> **Response R3:**
>
> We thank the reviewer for these suggestions and respond on three fronts.
>
> **Regarding TICA.** TICA plots on our generated trajectories compared with MDGEN and ProAR are provided in https://anonymous.4open.science/r/BioDynaSpec-4F51/figures/TICA.
>
> **Extended metrics.** We add metrics:
>
> - **Local dynamics fidelity**: PCA-PSD-L1 (L1 distance of power spectral densities in GT PCA modes); PCA-PSD-LogCorr (log-spectrum correlation in GT PCA modes); tICA-PSD-L1 (L1 distance of power spectral densities in GT tICA modes); tICA-PSD-LogCorr (log-spectrum correlation in GT tICA modes).
> - **Covariance consistency**: CFRE (relative Frobenius error of covariance matrices in the GT PCA subspace); Covariance-Eig-RRMSE (relative RMSE of covariance spectra in the GT PCA subspace).
>
> **Table S2.** Local dynamics fidelity and covariance consistency of generated trajectories.
>
> | Category | Metric | MDGEN | ProAR | BioDynaSpec (Ours) |
> |---|---|---:|---:|---:|
> | **Local Dynamics Fidelity** | PCA-PSD-L1 $\downarrow$ | 1.331 | 1.138 | **1.053** |
> | | PCA PSD LogCorr $\uparrow$ | 0.395 | 0.670 | **0.817** |
> |  | tICA PSD L1 $\downarrow$ | 1.501 | 1.375 | **1.275** |
> | | tICA PSD LogCorr $\uparrow$ | 0.380 | **0.662** | 0.661 |
> | **Covariance Consistency** | CFRE $\downarrow$ | 1.563 | 2.030 | **0.989** |
> | | Covariance Eig RRMSE $\downarrow$ |  1.184 | 1.625 | **0.941** |
>
> BioDynaSpec surpasses MDGEN and ProAR on PSD-based local-dynamics and covariance-error metrics (Table S2), capturing more accurate **spectral structure of near-equilibrium fluctuations** and matching to covariance geometry of the reference MD ensemble better.
>
> **Qualitative visualization.** Demos are at https://anonymous.4open.science/r/BioDynaSpec-4F51/demos/biodynaspec, showing generated dynamics trajectories vs. reference MD.

---

> > ### Author Rebuttal · Reviewer_Bk4K · 2026-04-03
> >
> > Thank the authors for the rebuttal. This is comprehensive and fully addressed my concerns. I am raising the score to Accept.

---

### Decision · Program_Chairs · 2026-04-30

**Decision:**

Accept (regular)

**Comment:**

This paper proposes a spectral-domain autoregressive method for protein dynamics trajectory generation in a coarse-to-fine manner, using harmonic-inspired structural bias to mitigate error accumulation seen in spatial models. Spectral domain framing is well-motivated for long-term MD trajectory generation and the manuscript is well-written with a grounded theoretical perspective.

**Rebuttal summary** Main concerns from reviewers include window size effects, boundary continuity, inference cost, and metric alignment with prior work. In their response, authors provided window size ablations, complexity analysis confirming cost-effectiveness, and additional validity metrics. However, we note the added metrics revealed suboptimal conformation quality and quality-motion trade-offs, along with other limitations acknowledged by the authors: CA-only representations, window boundary artifacts, and a focus on near-equilibrium dynamics.

**Recommandation: Accept** The conceptual contribution is valuable and the approach is well-grounded or mid-to-long-term trajectory generation, but empirical limitations and scope constraints should be clearly disclosed in the final version.